# Active Learning-Based Species Range Estimation

**Christian Lange[1]**     **Elijah Cole[2, 3]**     **Grant Van Horn[4]**     **Oisin Mac Aodha[1]**

[1]University of Edinburgh     [2]Altos Labs     [3]Caltech     [4]UMass Amherst

## Abstract

We propose a new active learning approach for efficiently estimating the geographic range of a species from a limited number of on the ground observations. We model the range of an unmapped species of interest as the weighted combination of estimated ranges obtained from a set of different species. We show that it is possible to generate this candidate set of ranges by using models that have been trained on large weakly supervised community collected observation data. From this, we develop a new active querying approach that sequentially selects geographic locations to visit that best reduce our uncertainty over an unmapped species' range. We conduct a detailed evaluation of our approach and compare it to existing active learning methods using an evaluation dataset containing expert-derived ranges for one thousand species. Our results demonstrate that our method outperforms alternative active learning methods and approaches the performance of end-to-end trained models, even when only using a fraction of the data. This highlights the utility of active learning via transfer learned spatial representations for species range estimation. It also emphasizes the value of leveraging emerging large-scale crowdsourced datasets, not only for modeling a species' range, but also for actively discovering them.

## 1   Introduction

Understanding the geographic range that a biological species occupies is a fundamental piece of information that drives models for ascertaining how vulnerable a species is to extinction threats [2], for quantifying how they respond to climate induced habitat change [29], in addition to being important for assessing biodiversity loss. Estimated range maps are typically generated from statistical or machine learning-based models that are parameterized from sparsely collected in situ observations [7]. Such in situ observations can be obtained through various means, e.g., via experts conducting detailed field surveys to record the presence or absence of a particular species in a defined geographic area [27], from community scientists that record incidental observations in a less structured manner [44], or from autonomous monitoring solutions such as camera traps [46]. One of the major limiting factors in scaling up the generation of reliable range maps to hundreds of thousands of species is the underlying collection of data.

Recently, community science-based platforms such as iNaturalist [4], eBird [44], and PlantNet [3] have proven to be an appealing and scalable way to collect species observation data by distributing the effort across potentially millions of participants around the world. However, due to the incidental nature of the observations collected on these platforms (i.e., users are not directed towards a specific geographic location and asked to survey it exhaustively for the presence of a species of interest), there can be strong biases with respect to the places people go and the types of species they report [11]. Additionally, a conflict exists in that the ranges of rare species are the hardest to characterize owing to the lack of data, but estimating their ranges would actually provide the most useful data for conservation purposes [30]. In this work, we explore an alternative collection paradigm for efficiently estimating geographic ranges of previously unmapped species inspired by ideas from

37th Conference on Neural Information Processing Systems (NeurIPS 2023).

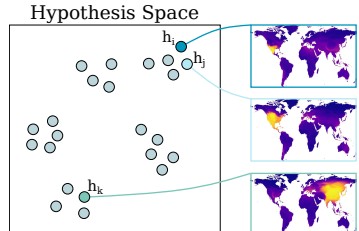
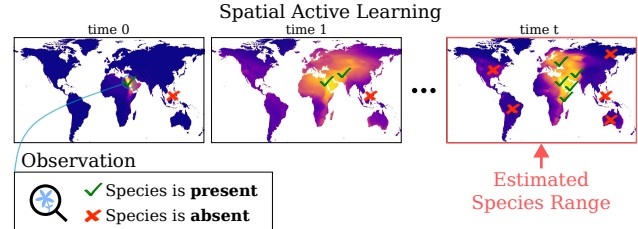

Figure 1: Our goal is to estimate the geographic range of a species (i.e., the locations where the species can be found during its lifetime) from a small number of actively selected in situ observations. (Left) We assume we have access to a hypothesis set of candidate species range estimation models, where each member of the set encodes the range for the species it was trained on. Intuitively, models that are close in this space encode similar spatial ranges (e.g., $h_i$ and $h_j$). (Right) At each time step in our active learning-based species range estimation paradigm, we sample a geographic location, a human then goes there to determine if the species is present or not, and then we update our predicted range based on the new data. The hypothesis space on the left is used to guide the selection of locations via our active querying approach.

active learning [41]. Our objective is to minimize the number of geographic locations that need to be sampled to accurately estimate a species' geographic range. To achieve this, we leverage recent advances in deep learning methods for joint species range estimation [13].

Existing work in species range estimation typically focuses on the offline setting whereby the entire set of observations is assumed to be available at training time [18; 12; 13]. As a result, the range estimation model is trained once offline and not updated. There have been attempts to use the trained models to predict which geographic location to investigate next to determine if a species might be present [38; 32]. However, these methods have typically not been developed, or quantified, in a fully online setting where new data is requested sequentially to update the model. Our approach instead makes the assumption that the geographic range of an unmapped species can be represented by a weighted combination of existing ranges. Through iterative active querying guided by these existing ranges, we efficiently estimate these weights online using only a small number of observations. An overview of this process is illustrated in Fig 1.

Our core contribution consists of a novel active learning-based solution for efficient species range estimation. We quantitatively evaluate the performance of our approach and compare it to alternative active learning methods on two challenging datasets containing expert-derived evaluation range data from one thousand species. We obtain a 32% improvement in mean average precision after ten time steps compared to a conventional active learning baseline on our globally evaluated test set. This translates to a significant reduction in the number of locations that need to be sampled to obtain a reliable estimate of a species range. Code for reproducing the experiments in our paper can be found at `https://github.com/Chris-lange/SDM_active_sampling`.

## 2   Related work

### 2.1   Species distribution modeling

Species distribution modeling refers to a family of methods that avail of species observation and geospatial data to estimate the geographic range of a species [17]. These models typically take some encoding of a spatial region of interest as input and output a numerical score indicating how likely it is for a particular species to be found there. There is a large body of work on this topic, but existing methods can be broadly categorized based on the type of data they train on (e.g., presence-absence or presence-only data) and the output quantity they attempt to predict (e.g., species presence or abundance) [7].

In the case of presence-absence data, at training time we have access to observations containing a set of locations where a species of interest is confirmed to be present and also locations where it has been confirmed to be absent. The advantage of this setting is that it is easy to fit standard supervised classification methods to the training data, e.g., decision trees [19]. This type of data

can be obtained from skilled observers that fill out detailed checklists recording the species they observe in a given area and time [44]. In contrast, in the presence-only setting, we do not have access to confirmed absences. Existing methods either factor this into the model [34] or synthetically generate *pseudo*-absence data for training [35]. Presence observations are easier to collect than presence-absence data as they are gathered opportunistically by observers [4]. Like all data-driven methods, these approaches have difficulties in estimating accurate ranges in the low data regime [30].

While traditional machine learning solutions have been shown to be effective for range estimation [18], recently we have observed a growing interest in the application of deep learning methods to the task [12; 8; 31; 16; 45; 13; 15]. In contrast to the majority of existing single species methods, these deep approaches jointly model multiple species in the same model, thus exploiting the fact that some species share similar range characteristics. In this work, we do not innovate on the underlying range estimation models used, but instead we show that existing methods that are trained on readily available weakly supervised presence-only data can provide transferable representations that can be used by active learning-based methods.

### 2.2 Active learning and species range estimation

Motivated by the fact that obtaining sufficiently diverse and informative training data can be one of the main bottlenecks when applying machine learning solutions to many real-world problems, active learning attempts to frame the data collection process as an optimization problem [41]. Through interaction with a user, the goal is to minimize the number of datapoints that need to be labeled while still obtaining high performance on the task of interest, e.g., in our case, estimating a species' range. A variety of active querying strategies have been proposed in the literature, including uncertainty sampling [28], committee-based methods [43], and methods that seek to label the datapoint that would have the largest impact on the model's parameters [42], among others.

The optimal design of spatial surveys for sampling a geographic region of interest when estimating spatially varying quantities is a central question in geostatistics [14] and has applications in tasks such as sensor placement [26]. Both non-adaptive and sequential methods have been explored for related tasks in the context of tractable parametric models [40]. In the case of deep learning [36], active sampling methods have also been shown to be effective for predicting spatially varying quantities in the context of remote sensing data [37]. However, the application of active sampling-based methods for large-scale species range estimation remains under explored. While we focus on estimating species ranges, our approach is applicable to any spatially varying quantity where it is possible to obtain a set of possible related predictions. Potential alternative applications include sensor placement, geographic priors for image classification, and disease modeling, in addition to related tasks such as active data collection for training image classifiers.

Xue et al. [48] explored different methods for guiding observers to under sampled locations to address the issue of spatial data bias in crowd collected datasets. However, their focus is not on estimating ranges, but instead to attempt to counteract data sampling biases. There have been attempts to use species distribution models to help guide the search for under observed species [25; 47; 23]. Rosner-Katz et al. [38] proposed a spatial sampling approach that involves stacking the outputs from a small number of range estimation models in order to test if locations that are estimated to contain rare species are good locations to check for other rare species. Marsh et al. [32] outlined an approach for defining a utility score for unsampled locations by measuring the difference in the output of a distribution model that is updated assuming the species of interest occurs there, or does not. This score can be used for suggesting locations to sample. However, it is expensive to compute as it requires updating the model for each possible location of interest. Different from these existing works, we propose a new efficient approach for the sequential acquisition of species presence or absence observations to efficiently parameterize models for range estimation.

## 3 Active species range estimation

### 3.1 Species range estimation

The goal of species range estimation is to predict the geographic range that a species occupies given a sparse set of observations at training time. Specifically, in the presence-absence setting [7], we are given a dataset $\mathcal{S} = \{(\boldsymbol{x}_i, y_i)\}_{i=1}^N$, where each $\boldsymbol{x}_i$ is a feature vector encoding a geographic location

of interest (i.e., a specific latitude and longitude). In this work we focus on spatial observations, but it is possible to extend our models to spatio-temporal data. The corresponding $y_i \in \{0, 1\}$ is a binary label indicating if the species has been observed to be present ($y_i = 1$) or found to be absent ($y_i = 0$) at the geographic location represented by $\boldsymbol{x}_i$.

The input features $\boldsymbol{x}$ can simply represent transformed latitude and longitude values (i.e., $\boldsymbol{x} = [lat, lon]$), a set of environmental features that encode information about the climate and/or habitat present at the location (i.e., $\boldsymbol{x} = [e_1, e_2, \ldots, e_d]$) [12], or could be a set of latent features extracted from a geospatially aware deep neural network (i.e., $\boldsymbol{x} = f([lat, lon])$) [31]. In our binary classification setting, the goal is to learn the parameters of a model $h$ such that it correctly classifies the observed data. This can be achieved by minimizing the cross-entropy loss with respect to the training data:

$$L_{CE} = -\sum_{(\boldsymbol{x},y) \in \mathcal{S}} y \log(h(\boldsymbol{x})) + (1 - y) \log(1 - h(\boldsymbol{x})). \tag{1}$$

Here, we represent a prediction from the model for one species as $h(\boldsymbol{x}) \in [0, 1]$. In the simplest case, $h$ could be a logistic regressor with $h(\boldsymbol{x}) = \sigma(\boldsymbol{\theta}^\top \boldsymbol{x})$, where $\sigma$ is the sigmoid function and $\boldsymbol{\theta}$ is the parameter (i.e., weight) vector that is learned for each species. Once trained, the model can be densely evaluated across all locations of interest (e.g., the entire globe) to generate a predicted range map for one species.

## 3.2 Active location sampling

In the previous section, we assumed we had access to a labeled training set $\mathcal{S}$ to train the range estimation model $h$. As noted earlier, obtaining this data on a global scale for rare or difficult to observe species can be extremely challenging as well as time consuming, even with a spatially distributed set of observers. Given this, it would be highly desirable to target limited observation resources to geographic locations that would result in observations that would be most informative for estimating the parameters of the model.

By framing our task as an active learning problem [41], our goal is to efficiently select and prioritize geographic locations for observation in order to improve the performance of the range estimation model for a given species while minimizing the total number of required observations. The active learning process can be broken down into three main steps; *querying/sampling*: determining the next location to observe, *labeling*: obtaining the observation label for the sampled location (i.e., is a species present or absent over a specified temporal interval), and *updating*: changing the model parameters based on the newly acquired data. Greedy active learning approaches simply query the next datapoint to be sampled one instance at a time. In this greedy setting, after a datapoint is observed and labeled, the model is updated. This process continues until some termination criteria is met, e.g., a maximum number of iterations or only a minor change in the model parameters is obtained.

One of the most commonly used strategies for greedy active learning is uncertainty sampling [28]. This involves sampling the feature that represents the location $\boldsymbol{x}$ that the current model is most uncertain about:

$$\boldsymbol{x}^* = \arg\min_{\boldsymbol{x}} |0.5 - P(y = 1 | \boldsymbol{x}, \mathcal{S}^t)|. \tag{2}$$

Here, $P(y = 1 | \boldsymbol{x}, \mathcal{S}^t) = h^t(\boldsymbol{x})$ indicates the output of the model at the time step $t$, and $\mathcal{S}^t = \{(\boldsymbol{x}_1, y_1), (\boldsymbol{x}_2, y_2), \ldots, (\boldsymbol{x}_t, y_t)\}$ is the set of training observations acquired up to and including time $t$. Once sampled, $\boldsymbol{x}^*$ is labeled (i.e., an individual goes to the location represented by $\boldsymbol{x}^*$ and determines if the species is present there, or not) and then added to the updated set of sampled observations $\mathcal{S}^{t+1} = \mathcal{S}^t \cup (\boldsymbol{x}^*, y^*)$. Finally, the current model is updated by fitting it to the expanded set of sampled observations. Later in our experiments, we compare to uncertainty sampling, in addition to other baseline active learning methods, and show that they are not as effective for the task of species range estimation in early time steps.

## 3.3 Leveraging candidate range estimation models

Unlike conventional classification problems found in tasks such as binary image classification, the ranges of different species are not necessarily distinct, but can instead have a potentially large amount of overlap (see Fig. 1 (left)). Inspired by this observation, we propose a new active sampling strategy that makes use of a set of candidate range estimation models.

We assume that we have access to a set of candidate pretrained models $\mathcal{H} = \{h_1, h_2, \ldots, h_k\}$. Each individual model encodes the spatial range for a different species, with one model per species. In practice, these can be a set of linear models (e.g., logistic regressors) that operate on features from a geospatially aware deep neural network, where each model has an associated weight vector $\boldsymbol{\theta}_k$. Having access to a large and diverse $\mathcal{H}$ may initially seem like a strong assumption, however in practice, it is now feasible to generate plausible range maps for thousands of species thanks to the availability of large-scale crowd collected species observation datasets [45; 13]. Importantly, we assume the species that we are trying to model using active learning is *not* already present in $\mathcal{H}$.

Assuming the candidate models are linear classifiers, we can represent our range estimation model $h$, for an unseen species, as a weighted combination of the candidate models' parameters:

$$\boldsymbol{\theta}^t = \sum_{k=1}^{|\mathcal{H}|} P(h_k|\mathcal{S}^t)\boldsymbol{\theta}_k. \tag{3}$$

By averaging the parameters of our candidate models we generate a compact linear classifier that can be used to predict the presence of the new species, without requiring access to all models in $\mathcal{H}$ during inference. When a new observation is added to $\mathcal{S}^t$, the posterior $P(h_k|\mathcal{S}^t)$ is updated based on the agreement between a model $h_k$ and the observed data in $\mathcal{S}^t$. Assuming a uniform prior over the candidate models in $\mathcal{H}$, we model the posterior as $P(h_k|\mathcal{S}^t) \propto P(\mathcal{S}^t|h_k)$, and the resulting likelihood is represented as:

$$P(\mathcal{S}^t|h_k) = \prod_{(\boldsymbol{x},y)\in\mathcal{S}^t} yh_k(\boldsymbol{x}) + (1-y)(1-h_k(\boldsymbol{x})). \tag{4}$$

This is related to the query-by-committee-setting in active learning [43], where our candidate models represent the individual 'committee' members. However, instead of being trained on subsets of the sampled data online during active learning, our candidate models are generated offline from a completely different set of observation data from a disjoint set of species.

### 3.3.1 Active learning with candidate range estimation models

Given our estimate of the model parameters from Eqn. 3, we can apply any number of active learning query selection methods from the literature to perform sample (i.e., location) selection. However, while our candidate models in $\mathcal{H}$ provide us with an effective way to estimate the model parameters online, we can also use them to aid in the sample selection step. Specifically, given the likelihood of an observation from a model, weighted by $P(h_k|\mathcal{S}^t)$, we choose the location $\boldsymbol{x}^*$ to query that is the most uncertain:

$$\boldsymbol{x}^* = \underset{\boldsymbol{x}}{\arg\min} \, |0.5 - \frac{1}{|\mathcal{H}|} \sum_{h_k \in \mathcal{H}} h_k(\boldsymbol{x})P(h_k|\mathcal{S}^t)|. \tag{5}$$

The intuition here is that we would like to sample the location that is currently most uncertain across all the candidate models. Our approach is motivated by the desire to make use of existing well characterized ranges when modeling the range of a less well described species, where observations may be very limited, to reduce the sampling effort needed.

The success of our approach hinges on the hypothesis set $\mathcal{H}$ containing a sufficiently diverse and representative set of candidate models. If the target test species' range is very different from any combination of the ones in $\mathcal{H}$, our ability to represent its range will be hindered. Additionally, even if the test species' range is broadly similar to some weighted combination of the models in $\mathcal{H}$, some finer-grained nuances of the range might be different. To address this, we extend the hypothesis set by adding an additional model $h_{\text{online}}$. $h_{\text{online}}$ is also a logistic regressor, but it is trained on the progressively accumulated observations from the test species that are obtained during active learning using a cross-entropy objective from Eqn. 1. Adding $h_{\text{online}}$ ensures that we also have access to the maximum likelihood model according to the observed data.

We refer to our approach as `WA_HSS`, which indicates that it uses a `Weighted Average` of the hypothesis set to represent the model weights, and performs `Hypothesis Set Selection` during active querying. Similarly, `WA_HSS+` is our approach with the inclusion of $h_{\text{online}}$.

# 4 Experiments

In this section we quantitatively evaluate our active learning approach on two existing benchmark datasets which we adapt to our online learning setting.

## 4.1 Implementation Details

**Experimental paradigm.** Our experiments are designed to compare different active learning strategies using synthetic presence-absence data drawn from expert-derived range maps. We begin by discretizing the world into a set of valid explorable locations which correspond to the centroids of all land overlapping resolution five H3 cells [1]. Here, each cell encompasses an area of 252 km$^2$ on average. At each time step, and for each active learning strategy, we select one location $x^*$ to query from any valid location that has not yet been sampled for the presence (or absence) of the species of interest. The corresponding species presence or absence label $y^*$ for $x^*$ is then generated according to the "ground truth" range map. In the real world, this step represents an observer visiting the queried location and searching for the species of interest. While our model could evaluate any continuous location, using H3 cell centroids ensures that queryable locations are equally spaced across the surface of the globe. We commence each experiment by randomly sampling one presence and one absence observation for each species. Experiments are performed independently for each species in the test set, and there is no interaction between species during the observation process. This is realistic, as in practice it would be unreasonable to assume that a single observer would be able to confirm the presence of more than a small number of species owing to their finite knowledge of different biological organisms. We repeat each experiment three times using different initial labelled examples and show standard deviations as error bars. We provide additional implementation details and a description of each of the baseline methods in the appendices.

**Training.** For our feature representation $x$ (discussed in Sec. 3.1), we use a learned spatial implicit neural representation obtained from a pretrained neural network. Specifically, we train a fully connected network $f$ with presence-only data using the $\mathcal{L}_{\mathrm{AN-full}}$ loss proposed in [13]. We use the same original set of publicly available data from iNaturalist [5] but retrain the model from scratch after removing species that are in our test sets. This results in 44,181 total species. The linear classifiers corresponding to these species form the set of candidate models $\mathcal{H}$ used by our approach. Each model in $\mathcal{H}$ has a dimensionality of 256, i.e., each is a logistic regressor with 256 weights. During the active learning phase, every time a labeled observation is obtained, for models that require a linear model to be estimated (e.g., LR models or models with $h_{\mathrm{online}}$), we update the classifier parameters $\theta$ for the species in question using the cross-entropy objective in Eqn. 1. We do not update the parameters of the feature extractor $f$, which are kept frozen. In later experiments, we quantify the impact of different feature representations by changing the feature extractor, e.g., by comparing models trained with environmental features as input or randomly initialized networks.

**Datasets.** We make use of two expert curated sources of range maps for evaluation and to generate labels for observations during the active learning process: International Union for Conservation of Nature (*IUCN*) [2] and eBird Status and Trends (*S&T*) [21]. We use the procedure outlined in [13] to preprocess the data and randomly select 500 species from each data source for evaluation. We apply an ocean mask to remove any cell that overlaps with the ocean to make the active learning procedure more "realistic" as we do not expect naturalists interested in improving a land based species range to explore the ocean to ensure that the species is not present there. This data represents our expert-grade presence-absence evaluation range map data. Unlike the *IUCN* data which has a confirmed presence or absence state for each location on the earth, the *S&T* data does not cover the entire globe. In the case of *S&T*, the valid region is different on a species by species case and is biased towards the Americas. As a result, during *S&T* evaluation, cells which are not included within the valid region for each species are excluded from sampling and evaluation. The distribution of each test set is illustrated in the appendices.

**Evaluation metrics.** Each experiment consists of comparisons between different active learning strategies where we start with the same initial set of two randomly selected observations (i.e., one absence and one presence). We report performance using mean average precision (MAP), where a model is evaluated at the centroid of each valid H3 cell (i.e., limited to cells that only overlap with land) and is compared to "ground truth" presence-absence obtained from the test set expert-derived

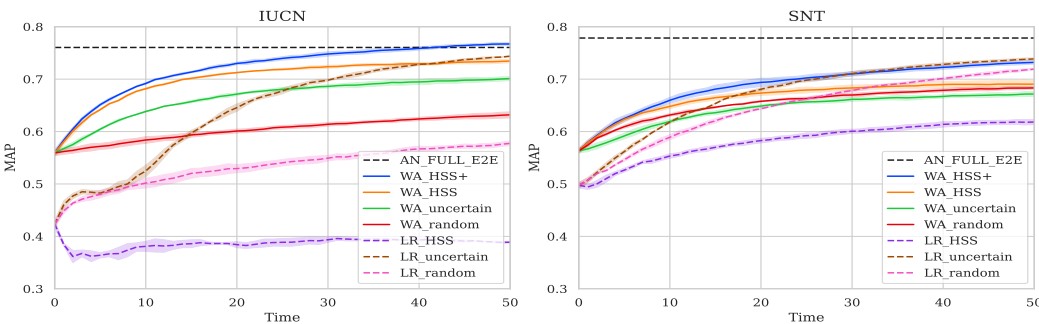

Figure 2: Comparison of different active learning querying methods for the task of species range estimation. For each of the two datasets, we display the performance of different active learning methods over time, where higher values are better. The results depict the average of three runs for each method. We observe that methods that use a weighted combination of the candidate models result in the best performance (e.g., `WA` versus `LR`) particularly in early timesteps. Our `WA_HSS+` approach performs best overall in both datasets. (Left) *IUCN* results, where we observe a large difference between methods. (Right) *S&T* results, where the performance difference is smaller but our approach is superior in the earlier timesteps.

range maps. During active learning, we measure average precision across all species in the test set at each time step and average them to obtain a single MAP score.

**Baselines.** In addition to evaluating our `WA_HSS+` approach, we compare to several baseline active learning strategies. These include `random` and `uncertainty` sampling. For each strategy, we evaluate models resulting from a weighted combination from the candidate set (`WA`) and conventional logistic regressors (`LR`). We also compare to `AN_FULL_E2E`, the end-to-end trained implicit network method from [13] using their $\mathcal{L}_{AN-full}$ loss trained with all available data for each species. While this model is trained only on presence-only data using pseudo-negatives, it has access to many more observations per species (i.e., a minimum of 50 presence observations per species) and is not restricted to only sampling data from land. Additional comparisons are provided in the appendices.

## 4.2 Results

**Comparison of active learning strategies.** In Fig. 2 we present quantitative results on the *IUCN* and *S&T* datasets, and show qualitative results in Fig. 4. We compare our `WA_HSS+` approach to multiple different baseline active learning methods and observe superior performance compared to each. The performance on individual species can vary a lot, but the average MAP across all test species is similar from run to run.

There is a clear advantage when using our active sample selection method (i.e., comparing `WA_HSS+` to random sampling in `WA_random`) on the *IUCN* test set. The addition of the online model estimate also is beneficial (i.e., comparing `WA_HSS+` to `WA_HSS`). Estimating model parameters as a weighted combination of the hypothesis set (i.e., `WA` methods) is significantly better than conventional logistic regression (i.e., `LR` methods). The performance improvement on the *S&T* test set is less dramatic. Here, even the random selection baseline performs well. This can possibly be attributed to the limited valid locations to explore for the species, the relative larger sizes, and the American focus of its ranges compared to the more globally distributed *IUCN* data. However, we still observe an advantage when using `WA` methods in the early time steps. As an additional baseline, we also compare to an end-to-end trained presence-only model from [13] which has access to significantly more observation data at training time. We can see that our `WA_HSS+` method approaches, and then exceeds, the performance of this baseline in fewer than 40 time steps for the *IUCN* species.

**Impact of the feature space.** Our approach makes use of a weakly supervised pretrained feature extractor $f$ to generate the input features $x$. Here, we explore different features spaces: `nn loc feats` are the standard deep network features from [13] that uses location only inputs, `nn env feats` are deep network features that use location and 20 additional environmental covariates (i.e., features that describe local properties such as amount of rainfall and elevation) from [20], and `nn rand feats` are from a randomly initialized and untrained version of the standard network. We also

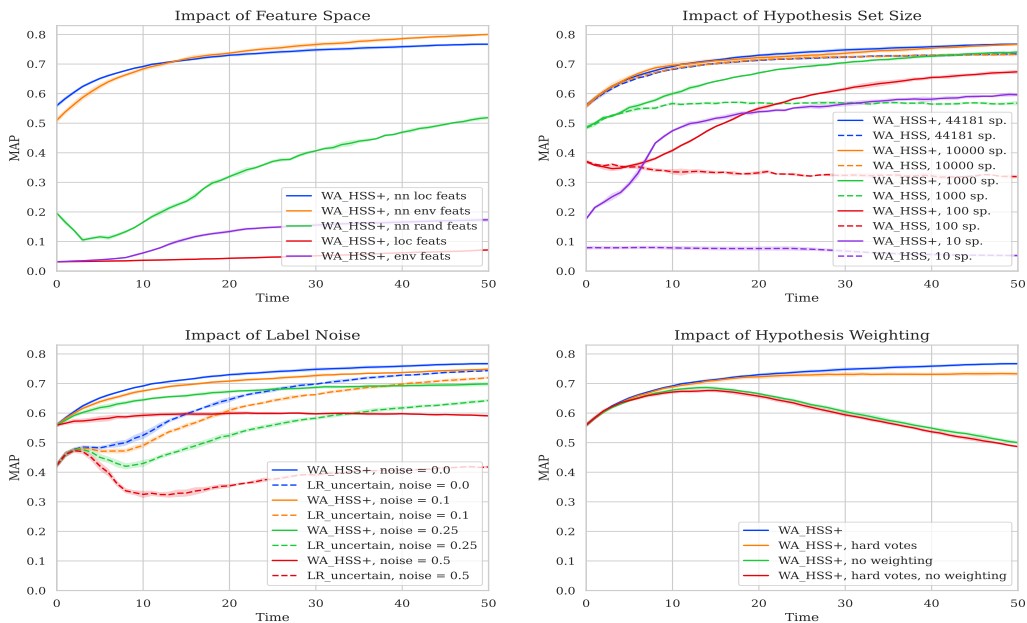

Figure 3: Ablations of our approach on the *IUCN* dataset. (Top Left) Impact of the feature space used to represent $x$. (Top Right) Impact of the number of species in the candidate set $\mathcal{H}$. (Bottom Left) Impact of observation noise by introducing false negatives, where higher label noise amounts indicate that the observer is more likely to incorrectly predict that a species is not present at a location. (Bottom Right) Impact of candidate model weighting during active query selection.

perform experiments using the same input features but without a deep feature extractor (i.e., `loc feats` and `env feats`). In this setting, the candidate models are just logistic regression classifiers operating directly on the input features. In Fig. 3 (top left) we observe that the deep feature extractors significantly outperform non-deep ones. This highlights the power of transfer learning from disjoint species in the context of range estimation. This has not been explored in the exiting literature, where conventional approaches typically train models on environmental covariate input features [18]. Interestingly, the fixed random higher dimensional projection of `nn rand feats` performs better than using only the low dimensional location or environmental features, `loc feats` and `env feats`. This is likely because the low dimensional features are not sufficiently linearly separable.

**Impact of number of candidate ranges.** One of the strengths of our approach is that it can leverage representations learned from disjoint species. In this experiment we quantify the impact of reducing the number of species available when training the backbone feature extractor which also results in a reduction of the size of the candidate set $\mathcal{H}$. Specifically, we train different feature extractors from scratch, where each model has access to a different subset of the data during training, e.g., `WA_HSS, 1000 sp` is trained on data from only 1,000 species. In Fig. 3 (top right) we unsurprisingly observe a drop in MAP when the size of $\mathcal{H}$ is small (e.g., < 1,000 species). As the size increases, the model performance improves. This illustrates the benefit of larger and more diverse candidate sets. This reduction has a larger impact on `WA_HSS` compared to `WA_HSS+` due to the inclusion of $h_{\text{online}}$ in the case of the latter.

**Impact of observation noise.** Here we simulate the impact of observation noise by exploring the setting where the species is actually present at the location but the observer misses it. Thus we obtain false negative observations, but not false positives. We simulate different noise rates where the observer misses the species at each time step with a probability sampled from {0%, 10%, 25%, 50%}. In Fig. 3 (bottom left) we observe that our approach is surprisingly robust to non-trivial amounts of observation noise, and only starts to fail when the noise level becomes very large (i.e., 50%). In contrast, the best logistic regressor baseline is much more impacted by such label noise.

**Impact of hypothesis weighting.** Unlike traditional query by committee methods where each committee member is trained on a subset of the available data [43], our hypotheses represent models

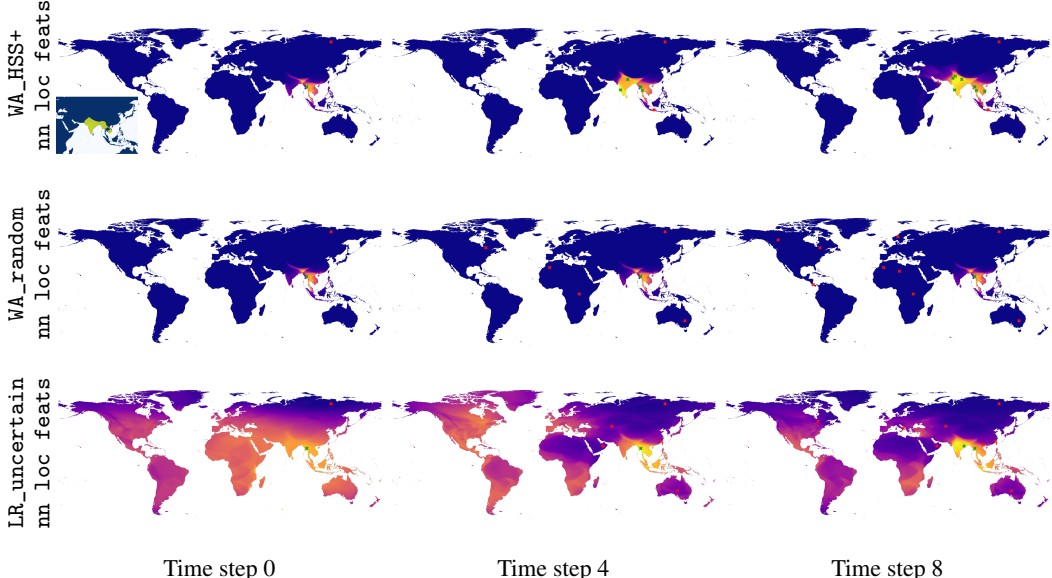

Figure 4: Predicted range maps for the `Yellow-footed Green Pigeon` for three different active learning strategies, illustrated over three different time steps. We also display the expert range map centered on India and Southeast Asia, inset in the top left. The queried locations are marked with ×, with green representing present locations and red representing absent ones. `WA_HSS+` quickly identifies both the Indian and Southeast Asian part of the species' range.

trained on *different* species. Thus, no individual model will necessarily have the same range as the test species. To compensate for this, we weight each $h \in \mathcal{H}$ by the likelihood of that hypothesis, $P(h|\mathcal{S}^t)$, as indicated in Eqn. 5. In Fig. 3 (bottom right) we observe that removing this weighting term (`no weighting`) significantly reduces the performance of `WA_HSS+`, particularly at later time steps where $\mathcal{S}^t$ contains enough samples to effectively identify and downweight hypotheses that are no longer consistent with the data. On the other hand, averaging the "soft votes" of each hypothesis (i.e., multiplying by $h_k(\boldsymbol{x})$ as in Eqn. 5) (`WA_HSS+`), or averaging the "hard votes" (i.e., $[\![h_k(\boldsymbol{x}) > 0.5]\!]$ used in `WA_HSS+ hard labels`) as in traditional query by committee methods [24], results in similar performance. The uncertainty of any one model is not particularly important, potentially due to the large number of hypotheses that can contribute to sample selection.

## 5   Limitations

In spite of the significant improvement in performance our approach provides, it does have some limitations which we leave for future work. Firstly, all active learning strategies we consider query single points rather than diverse batches [9]. This might limit the use of our strategies in situations where there are multiple observers available to gather data simultaneously. The geographic accessibility of specific locations is another limitation as we currently do not consider the practical difficulties associated with reaching certain geographic areas, e.g., remote regions. We also do not explore other forms of observation noise outside of failing to detect the species of interest when it is present. However, errors associated with misidentifying species are mitigated on platforms such as iNaturalist [4] who use a community consensus mechanism to assign labels to observations.

The performance of our of approach is limited by the expressiveness of the provided learned feature space and the diversity of species present in the candidate hypothesis set. We are also impacted by any inaccuracies in the candidate ranges that may result from their inability to capture very high spatial resolution range characteristics, in addition to any class imbalance or geographic sampling biases in the data used to train the underlying models. However, our method will further benefit from advances in handling noise and bias in the context of training joint range estimation models [11], in addition to improvements in architectures and location encodings [39]. Finally, the expert-derived evaluation data we use represents the best available source, but it is still not guaranteed to be free from errors as characterizing a species' true range is a challenging problem.

**Broader impacts.** The models trained as part of this work have not been validated beyond the use cases explored in the paper. Thus care should be taken when using them for making any decisions about the spatial distribution of species. We only performed experiments on publicly available species observation data, and caution should be exercised when using these models to generate range maps for threatened species.

# 6 Conclusion

Obtaining a sufficient number of on the ground observations to train species range estimation models is one of the main limiting factors in generating detailed range maps for currently unmapped species. To address this problem, we presented one of the first investigations of active learning in the context of efficiently estimating range maps from point observation data. We also proposed a new approach that makes use of a pre-existing learned environmental feature representation and a set of range maps from disjoint species to enable efficient querying of locations for observation. Even though this feature representation is derived from a model trained on weakly labeled presence-only data, we show that it is still very effective at transferring to new species. Our approach is computationally efficient as all of our operations are linear with respect to the input features which ensures that we can easily scale to tens of thousands of species. Through extensive quantitative evaluation, we demonstrated that it is possible to obtain a significant improvement in the quality of the resulting range maps with much fewer observations when compared to existing active learning methods.

**Acknowledgements.** This project was funded by the Climate Change AI Innovation Grants program, hosted by Climate Change AI with the support of the Quadrature Climate Foundation, Schmidt Futures, and the Canada Hub of Future Earth. Funding was also provided by the Cornell–Edinburgh Global Strategic Collaboration Awards.

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

# Appendix

## A  Additional comparisons

In Table A1 we report the numerical performance of our approach compared to multiple different active learning strategies, including additional ones that are not present in the main paper. Performance is reported using per-time step mean average precision (MAP), in addition to area under the MAP curve (MAP-AUC). We also illustrate a subset of these methods in Fig. A1. We observe that our `WA_HSS+` approach results in superior performance.

Table A1: Numerical results on the *IUCN* dataset reported at the end of four different time steps (5, 10, 20, or 50). Each row is a different active querying method. We report results as the average of three different runs.

|  | Time step 5 | | Time step 10 | | Time step 20 | | Time step 50 | |
|---|---|---|---|---|---|---|---|---|
|  | MAP | MAP-AUC | MAP | MAP-AUC | MAP | MAP-AUC | MAP | MAP-AUC |
| `WA_HSS+` | **0.65** | **0.61** | **0.69** | **0.64** | **0.73** | **0.68** | **0.77** | **0.72** |
| `WA_uncertain+` | 0.61 | 0.58 | 0.64 | 0.60 | 0.68 | 0.63 | 0.73 | 0.68 |
| `WA_random+` | 0.57 | 0.57 | 0.58 | 0.57 | 0.60 | 0.58 | 0.64 | 0.61 |
| `WA_positive+` | 0.58 | 0.57 | 0.58 | 0.58 | 0.58 | 0.58 | 0.58 | 0.58 |
| `WA_HSS` | 0.64 | **0.61** | 0.68 | 0.63 | 0.71 | 0.67 | 0.73 | 0.70 |
| `WA_uncertain` | 0.60 | 0.58 | 0.64 | 0.60 | 0.67 | 0.63 | 0.70 | 0.66 |
| `WA_random` | 0.57 | 0.57 | 0.58 | 0.57 | 0.60 | 0.58 | 0.63 | 0.60 |
| `WA_positive` | 0.58 | 0.57 | 0.58 | 0.57 | 0.58 | 0.58 | 0.59 | 0.58 |
| `LR_HSS` | 0.36 | 0.38 | 0.38 | 0.38 | 0.38 | 0.38 | 0.39 | 0.39 |
| `LR_uncertain` | 0.48 | 0.47 | 0.52 | 0.49 | 0.65 | 0.54 | 0.74 | 0.64 |
| `LR_random` | 0.48 | 0.46 | 0.50 | 0.48 | 0.53 | 0.50 | 0.58 | 0.53 |
| `LR_positive` | 0.39 | 0.40 | 0.43 | 0.41 | 0.48 | 0.43 | 0.50 | 0.47 |
| `LR_QBC` | 0.49 | 0.47 | 0.53 | 0.49 | 0.65 | 0.54 | 0.75 | 0.64 |
| `LR_EMC` | 0.47 | 0.46 | 0.51 | 0.47 | 0.61 | 0.52 | 0.71 | 0.62 |

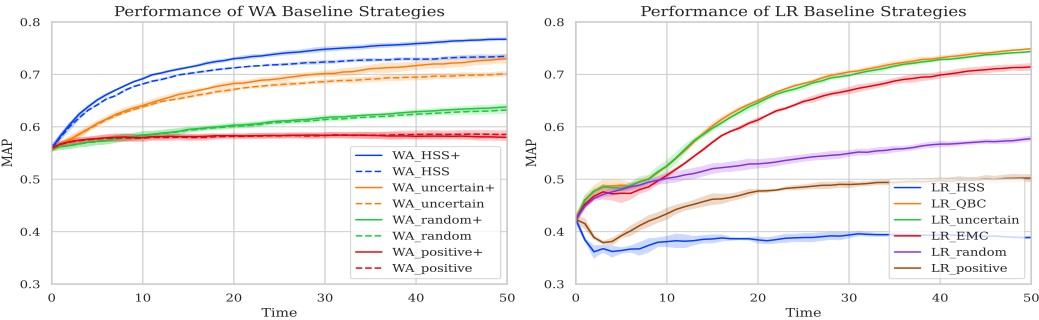

Figure A1: Comparison of additional active learning methods for the task of species range estimation on the *IUCN* dataset. We display the performance of different active learning methods over time, where higher values are better. This figure is similar to Fig. 2 (left) in the main paper, but includes additional comparisons.

## B  Visualizations

### B.1  Learned representations

In Fig. A2 (left), similar to the visualizations in [13], we display a visualization of the learned features from a neural network backbone. We use ICA to project the 256D vector to 3D and only display non-ocean features. Fig. A2 (right), we show a 2D TSNE visualization of the candidate model hypothesis set $\mathcal{H}$, where each point represents a different species weight vector $\theta$. We show the resulting range predictions (i.e., $\sigma(\boldsymbol{\theta}_i^\top \boldsymbol{x})$) for three different species ($h_1$ = Ludwig's Bustard, $h_2$ = Namib Rock Agama, and $h_3$ = Black-headed Ibis), where we observe that the two species with similar ranges ($h_1$ and $h_2$) have similar low-dimensional embeddings.

In Fig A3 we illustrate the learned combination of species' weight vectors that are used to generate predictions for three unseen test species. We observe that highly weighted species from $\mathcal{H}$ often form distinct clusters in the TSNE representation, where each cluster is comprized of species whose ranges form one part of the test set species range. This helps explain why it is meaningful to learn a linear combination of pretrained ranges as in our method. With very few observations species with similar ranges can be identified and exploited to create a good range map for the species of interest.

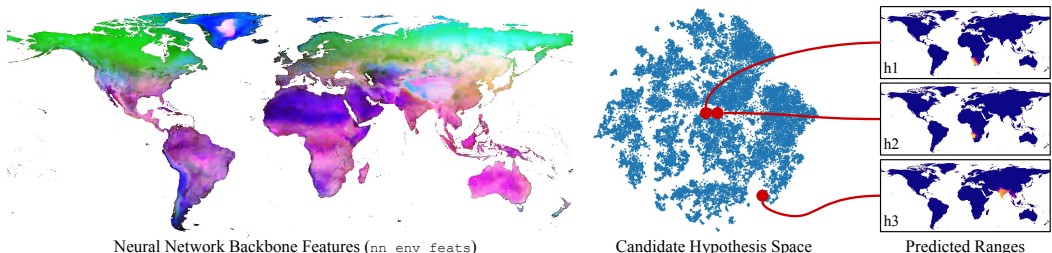

Figure A2: (Left) 3D ICA projection of features for an `nn env feats` neural network backbone. (Right) 2D TSNE visualization of $\mathcal{H}$ for a `nn coord feats` neural network backbone with predicted range maps shown for 3 species. Here, $h_1$ = Ludwig's Bustard, $h_2$ = Namib Rock Agama, and $h_3$ = Black-headed Ibis. Points close in the hypothesis space represent similar ranges.

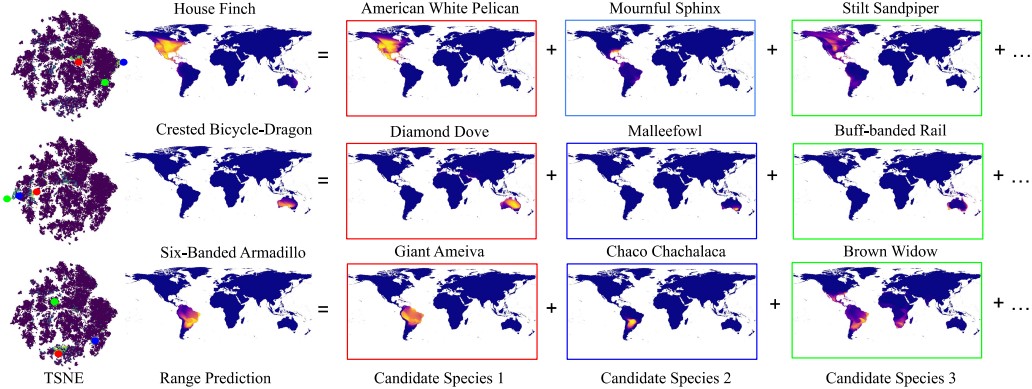

Figure A3: Visualization of the learned combination of species' weight vectors that are used to generate predictions for three unseen test species, each shown on a different row. On the left we show a TNSE visualization of the hypothesis set, where brighter colours indicate species that have a greater weight at the end of active learning for our `WA_HSS+` approach. For each of the species visualized, we indicate the highest weighted candidate species from a set of highly weighted locations that are indicated using a red, blue, or green marker on the left.

## B.2 Additional qualitative results

In Figs. A4 to A6 we display additional qualitative results. In each figure, we compare the outputs of different active sampling strategies across different time steps for the same species. In general, we observe that our `WA_HSS+` approach results in predictions that are closer to the expert range map (shown in the top left of each figure). However, Fig. A4, illustrates an interesting failure case where our approach fails to predict the species presence outside the region of the initial presence observation. However, from Fig. A11 we can see that the predictions for most species results in higher MAP scores for `WA_HSS+`.

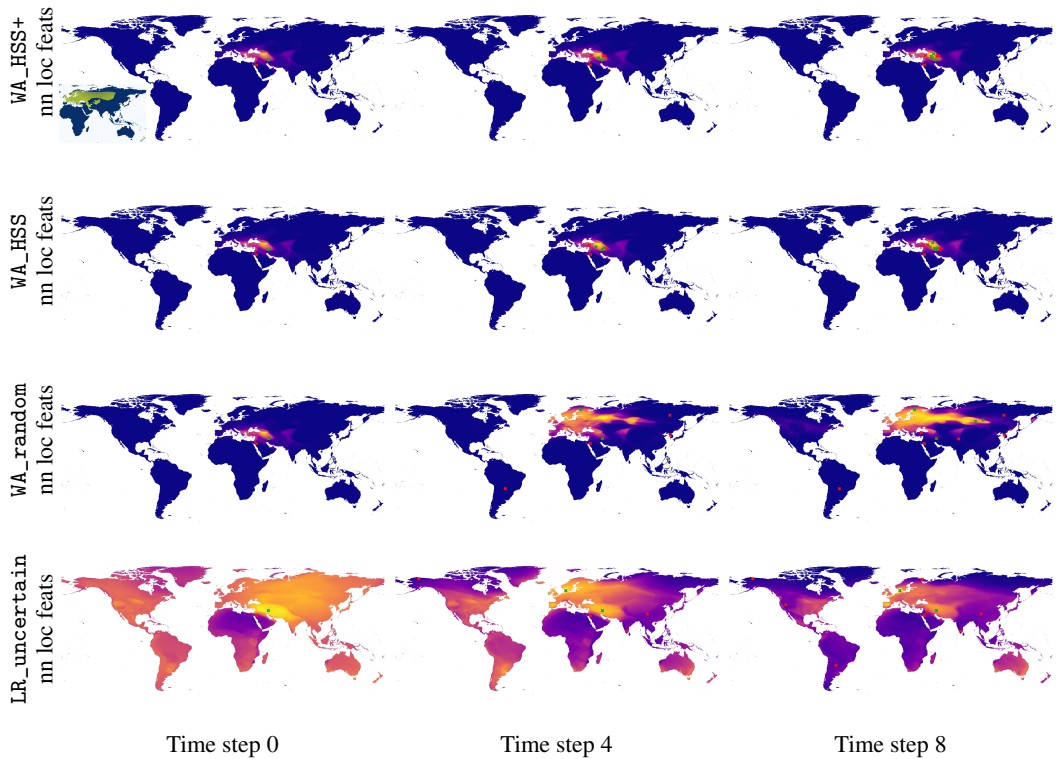

Time step 0       Time step 4       Time step 8

Figure A4: Predicted range maps for the `Yellowhammer` for four different active learning strategies, illustrated over three different time steps. We also display the expert range map, inset in the top left. The queried locations are marked with Xs, with green representing present locations and red representing absent ones. In this example `WA_HSS+` and `WA_HSS` fail to identify the most of the range while other sampling strategies are more successful.

## C   Implementation details

### C.1   Dataset details

In Sec. 4.1 in the main paper we noted that the *S&T* dataset does not span the entire globe and is more spatially biased compared to the *IUCN* dataset. In Fig. A7 we visualize the number of coarse locations that have a species presence for both datasets. We can see that the *IUCN* dataset on the left is more globally distributed, compared to the *S&T* dataset on the right. In addition, in the case of the *S&T* dataset, it is only possible to evaluate on a subset of possible locations as opposed to the *IUCN* dataset where we can evaluate on all possible locations on the earth's surface. In Table A2 we summarize the statistics of both datasets.

Table A2: Comparison of dataset statistics. We evaluate our models at a H3 [1] resolution of five, which results in 2,016,842 possible cells distributed across the globe before we exclude ocean locations. The *S&T* dataset covers a smaller percentage of the globe compared to the *IUCN* dataset.

|  | *IUCN* | *S&T* |
|---|---|---|
| Avg. # valid H3 cells | 569,094 | 162,008 |
| Avg. # positive H3 cells | 19,614 | 42,429 |
| Avg. # valid percent of globe | 28.22 | 8.03 |
| Avg. # positive percent of globe | 0.97 | 2.01 |
| Avg. # positive / valid percent | 3.45 | 26.19 |

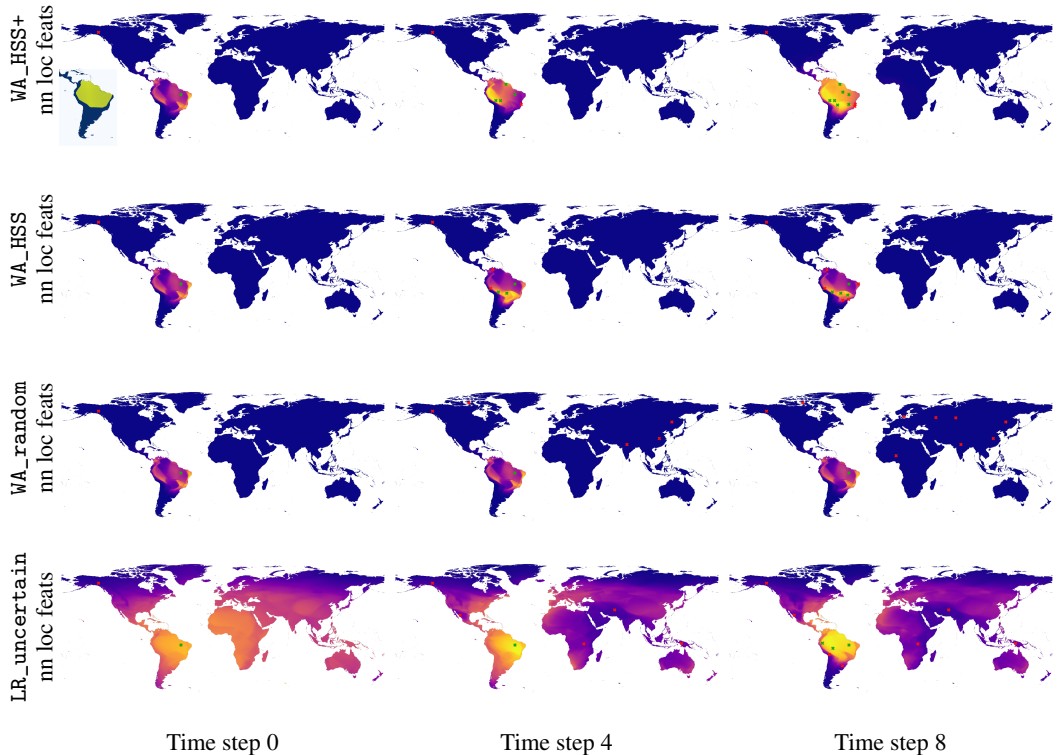

Figure A5: Predicted range maps for the `Gold Tegu` for four different active learning strategies, illustrated over three different time steps. We also display the expert range map centered on South America, inset in the top left. `WA_HSS+` quickly identifies the majority of the range.

## C.2 Input features

In our experiments we compare different input feature encodings $x$ to represent locations on the earth's surface. Unless otherwise specified, we use a four dimensional trigonometric encoding of the scaled latitude and longitude (either directly or via a neural network-based feature extractor). This encoding is used to prevent discontinuities at the connected edges of the world map [31]:

$$x = \left[ \sin\left(\frac{lat\pi}{90}\right), \cos\left(\frac{lat\pi}{90}\right), \sin\left(\frac{lon\pi}{180}\right), \cos\left(\frac{lon\pi}{180}\right) \right]. \tag{6}$$

In the results in Fig. 3 (top left) in the main paper we compare to models that use features derived from a neural-network backbone (i.e., those with `nn` in the name). If a neural network backbone is not specified, the input features are instead passed directly to the classifier. These neural network feature extractors are either randomly initialized using location only features and not trained (i.e., `rand`), or are trained using the procedure outlined in [13] with only location features (i.e., `loc`) as input or location and environmental covariates (i.e., `env`).

Environmental covariates are commonly used for species distribution models as it allows them to attempt to learn the "fundamental ecological niche" of the species of interest [7]. By also including the encoded coordinates we make our models spatially explicit and allow them to distinguish between locations that have similar environmental characteristics but are spatially distant. We use 20 environmental covariates (including elevation) generated from elevation and bioclimatic rasters at a five arc minute spatial resolution, taken from the widely used Worldclim 2.1 dataset [20]. Each covariate is normalized by subtracting the mean and dividing by the standard deviation, and NaN values are replaced with 0. Bilinear interpolation is used to generate corresponding environmental covariates for any given location. Models that take environmental features as input therefore accept 24 dimensional inputs, with four dimensions encoding location information and the remaining 20 representing the bilinearly interpolated environmental covariates at the location.

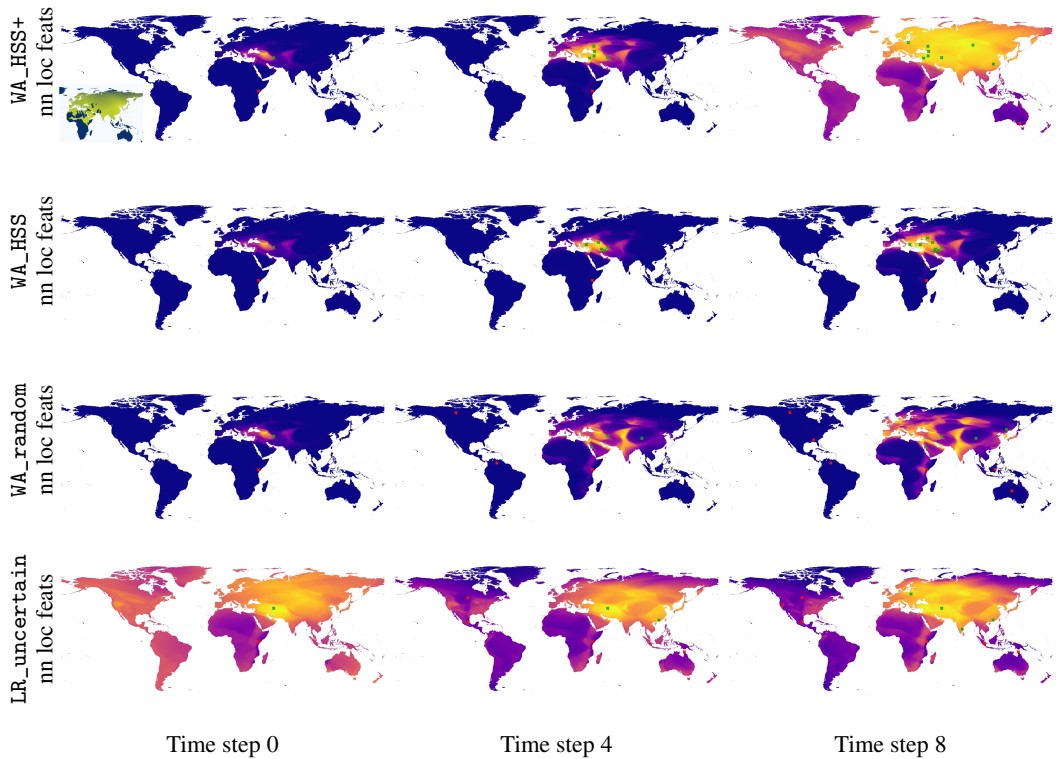

Figure A6: Predicted range maps for the `White Wagtail` for four different active learning strategies, illustrated over three different time steps. We also display the expert range map centered on Asia, Africa and Europe, inset in the top left. `WA_HSS+` quickly identifies the majority of the range.

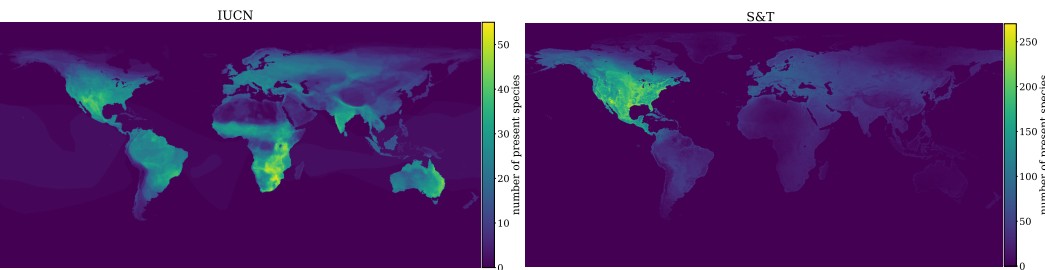

Figure A7: Count of the number species presences for each coarse geographic location for the *IUCN* (left) and *S&T* (right) datasets. We can observe that the *IUCN* dataset is much more globally distributed and less North American biased than the *S&T* dataset.

### C.3    Backbone feature extractor

In addition to the input encoding, our method involves extracting features from a deep species range estimation model trained on community collected presence-only data. This network uses the same architecture, training procedure, and hyperparameter settings as the $\mathcal{L}_{\mathrm{AN-full}}$ model from [13]. We also use the same training data from iNaturalist [5], but we remove classes (i.e., species) that are present in the full set of species (i.e., not just the 1,000 in our subset of the test species) in either of our test datasets. This reduces the number of species to 44,181.

The deep species range estimation model is a fully connected neural network consisting of an input layer, then four residual layers, each comprised of two fully connected layers with skip connections between them, and finally a classification head consisting of a linear classifier per-species, each of which predicts the presence or absence of a single species using a sigmoid output. It can be considered a location encoder $f$ that feeds into set of single layer classifiers $\mathcal{H}$ which all share the

same input features. ReLU nonlinearities are used between layers, and each hidden layer consists of 256 neurons. Dropout layers are also utilized. This model is trained for 10 epochs using Adam optimizer with a dropout probability of 0.5 and a batch size of 2048. Training the feature extractor takes 2.5 hours on an NVIDIA RTX A6000 GPU with 48GB of RAM.

After training, the frozen network $f$ can then be used as a feature extractor, generating 256 dimensional embeddings used for hypothesis set selection and for training new classifiers. Each per-species classifier $h$ can be used as the set of hypotheses $\mathcal{H}$ in our `WA` approaches. As $f$ is frozen during active sampling and the locations that can be sampled are limited to the centroid of resolution five H3 cells, features can be generated ahead of time by applying the feature extractor to each centroid and storing the results. Thus $f$ is not required during the active sampling process.

## C.4   Active sampling

In each experiment, we apply the chosen active sampling strategy on 500 held-out test species randomly selected from each of the International Union for Conservation of Nature (*IUCN*) [2] and eBird Status and Trends (*S&T*) [21] datasets. We use the same 500 species from each dataset for each experiment, and use the same initial samples consisting of a single presence and absence observation for each experiment. The species set in each dataset is not the same, e.g., the *S&T* dataset contains only birds. Each experiment is repeated three times with different seeds and initial samples. The same frozen feature extractor $f$ is used across all experiments except 'Impact of Feature Space' experiments in Fig. 3 in the main paper and the additional ablations in this appendix.

We evaluate our performance using per-time step mean average precision (MAP), and using area under the MAP curve (MAP-AUC). As we synthetically generate data from the underlying expert range maps provided, we do not have a specified train test split and merely evaluate at each valid H3 cell centroid and measure performance as the difference between a model's predictions and the expert-derived range map. Unlike [13], we evaluate our models only on land and not also over the ocean.

The logistic regression range estimation model $h$ we optimize during active sampling is implemented using `scikit-learn`'s `linear_model.LogisticRegression` class with default settings [33]. If the features from $f$ have been stored ahead of time, then each experiment can be performed without a GPU as it consists only of training and evaluating a logistic regression model. A single experiment using our `WA_HSS+` method involving 500 species across 50 time steps takes 4 hours using unoptimized code on an AMD EPYC 7513 32-Core Processor.

## C.5   Baseline strategies

Here we provide additional information about the baseline active sampling strategies we evaluated. In all cases, the set of possible locations from which to select a sample $x^*$ consists of all valid resolution five H3 cell centroids (i.e., those that are located over land and have not yet been sampled, and for the *S&T* dataset only those where the ground truth is provided). The `LR` strategies below update the range estimation model at each time step by minimizing the cross entropy loss in Eqn. 1 from the main paper.

**LR_random.** $x^*$ is selected randomly from all valid H3 cell centroids.

**LR_uncertain.** $x^*$ is determined by Eqn. 2, and corresponds to the location where the model's probability of presence is closest to 0.5. This is one of the most well established and commonly used methods of active sampling for classification.

**LR_EMC.** $x^*$ is selected to maximize the absolute gradient of the model parameters $\theta$ with respect to the loss function, taking an expectation over the current distribution of class labels. We are looking for the sample that will result in the largest expected model change. The intuition is that we expect this location will cause our model to "learn" the most [10].

**LR_HSS.** $x^*$ is selected according to Eqn. 5 in the main paper, where the weights of the hypothesis set $\mathcal{H}$ are determined by agreement between the hypotheses and the sampled data, and the location that the weighted hypotheses are most uncertain about is then selected. However unlike `WA_HSS`, $h$ is not updated by averaging these hypotheses, but by updating the weights for a logistic regression classifier to minimize the cross entropy loss.

**LR_QBC.** This 'query by committee' method [43; 24] is similar to `LR_HSS` above but $\mathcal{H}$ is now a set of classifiers trained on subsets of the currently available data $\mathcal{S}^t$ as in [6]. Due to the small size of $\mathcal{S}^t$, especially in early time steps, subsets are created by removing a single data point from $\mathcal{S}^t$, and then discarding subsets that only contain points from a single class.

**LR_positive.** $\boldsymbol{x}^*$ is selected as the location that has the highest probability of presence according to $h$. This querying method has been used within the ecology literature as a method for evaluating generated species distribution models or as a 'guided search' method [22; 23].

We also compare our `WA_HSS+` strategy to others that use weighted averaging of $\mathcal{H}$ to update $h$, but do not use hypothesis set selection to determine $\boldsymbol{x}^*$. The following `WA` strategies can also be modified by the inclusion of $h_{\text{online}}$, leading to **WA_random+**, **WA_uncertain+**, and **WA_positive+** baselines.

**WA_random.** $\boldsymbol{x}^*$ is selected randomly from all valid H3 cell centroids.

**WA_uncertain.** $\boldsymbol{x}^*$ is determined by Eqn. 2 in the main paper, and corresponds to the location where the model's probability of presence is closest to 0.5.

**WA_positive.** $\boldsymbol{x}^*$ is the location that has the highest probability of presence according to $h$.

# D   Additional ablations

Here we provide the results of additional ablation experiments.

## D.1   Impact of label noise

In Fig. A8 we demonstrate the impact of adding label noise for `WA_HSS+` compared to `WA_HSS`. This expands on the results in Fig. 3 (bottom left) in the main paper. While `WA_HSS+` outperforms `WA_HSS` in the presence of modest label noise, as more noise is added, the `WA_HSS` strategy performs slightly better than `WA_HSS+`.

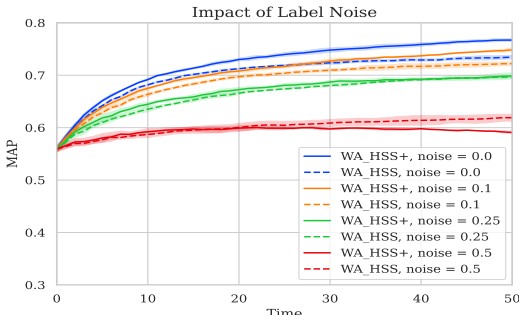

Figure A8: Impact of adding increasing amounts of label noise on the *IUCN* dataset. Our `WA_HSS+` and `WA_HSS` approaches are robust to label noise compared to conventional logistic regression-based uncertainty sampling used in `LR_uncertain` and seen in Fig. 3 (bottom left) in the main paper.

## D.2   Impact of hypothesis set size

In Fig. A9 (left) we illustrate the impact of removing species from the training set when training the backbone feature extractor. We compare the performance of `WA_HSS` using different backbone models trained with a limited number of classes to `WA_HSS` where we remove classifiers from $\mathcal{H}$ but keep the same feature extractor trained on all classes. Fig. 3 (top right) in the main paper shows a large impact from using backbones trained with fewer classes. Here we observe that in almost all cases the model using a backbone trained on all classes performs best, and the performance difference increases when more classes are removed.

In Fig. A9 (right) we further show how using backbones trained on fewer classes impacts the baseline active sampling strategy. The change in performance for the `LR_uncertain` strategy underlines that jointly modeling many species in the backbone is vital for good performance due to the features

extracted becoming more useful for range estimation, even when the predicted species ranges are not directly used, as they are in `WA_HSS`.

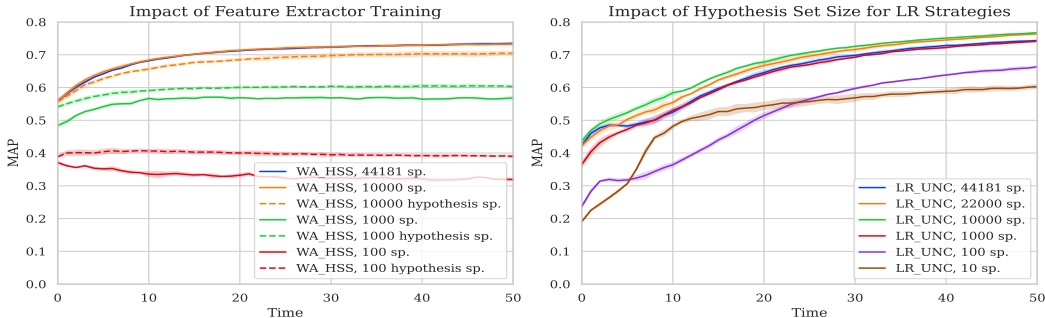

Figure A9: (Left) Impact of removing species from the backbone on the *IUCN* dataset. Here we compare models trained without species in the neural network backbone and hypothesis set (solid line) to models that use all the species to train the backbone but simply remove some from the hypothesis set (dashed lines). The solid lines are the same as the solid entries in Fig. 3 (top right) in the main paper. (Right) Impact of changing the hypothesis set size on the *IUCN* dataset for the `LR_uncertain` strategy. When many fewer species are used to train the backbone the performance is worse.

### D.3    Impact of feature space

In Fig. A10 we explore the impact of using learned input features for our `WA_HSS+` method and the logistic regression baseline `LR_uncertain`. The feature spaces are a subset of those shown in Fig. 3 (top left) from the main paper. We observe that using the features generated by the deep feature encoder (`nn loc feats`) is vital for good performance for both the `LR_uncertain` and `WA_HSS+` strategies. In contrast, only using location (`loc feats`) or location and environmental features (`env feats`) directly with no neural network results in poor performance.

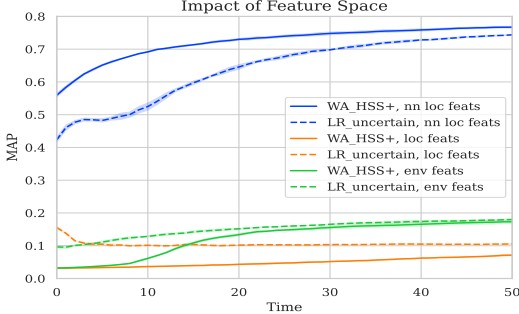

Figure A10: Impact of using different feature spaces evaluated on the *IUCN* dataset. Both methods require the use of the deep feature encoder features, `nn loc feats`, for good performance.

### D.4    Per-species performance

In Figs. A11 to A14 we display histograms of the per-species performance across six different time steps (2, 5, 10, 20, 30, and 50) on the *IUCN* dataset. Each individual histogram displays the distribution of MAP scores for each of the species in the *IUCN* dataset. In general, we observe a trend that performance improves over time, i.e., the mass of the histograms move the right. Our `WA_HSS+` approach in Fig. A11 is noticeably better than the naive random selection using logistic regression baseline in Fig. A14.

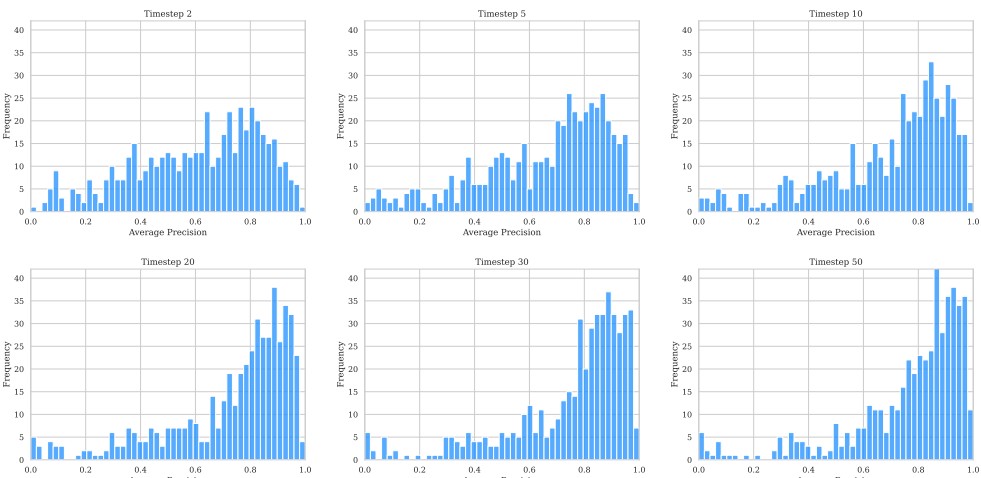

Figure A11: Histograms of per species MAP scores for our `WA_HSS+` approach across six different time steps on the *IUCN* dataset.

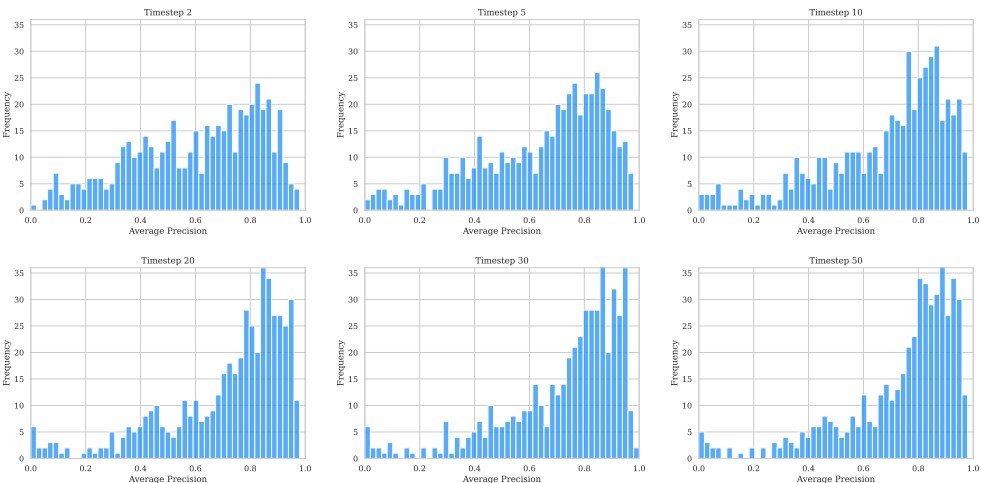

Figure A12: Histograms of per species MAP scores for our `WA_HSS` approach across six different time steps on the *IUCN* dataset.

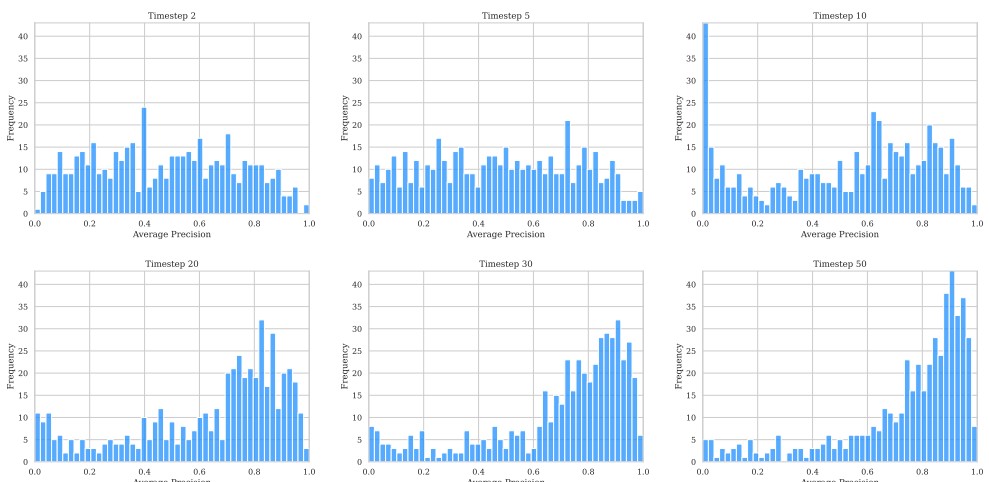

Figure A13: Histograms of per species MAP scores for `LR_uncertain` (i.e., logistic regression with uncertainty-based sample selection) across six different time steps on the *IUCN* dataset.

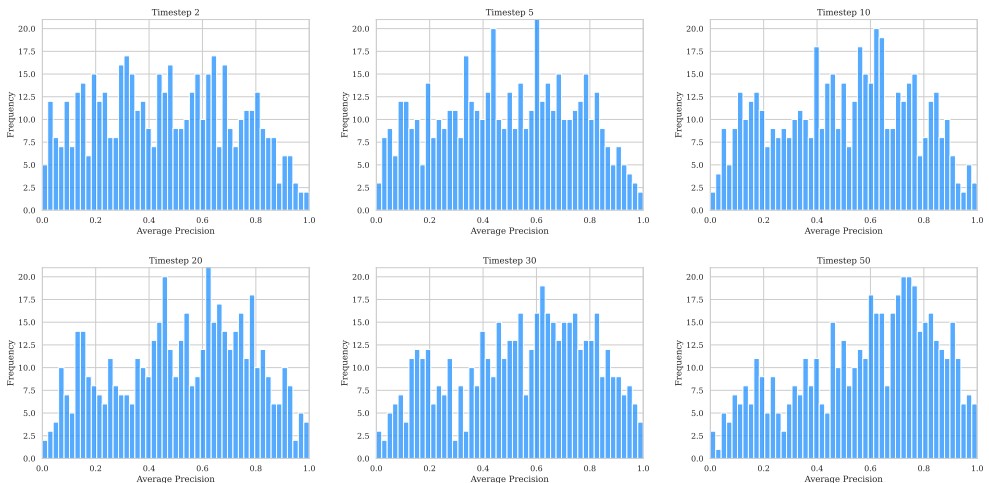

Figure A14: Histograms of per species MAP scores for the `LR_random` baseline (i.e., logistic regression with random sample selection) across six different time steps on the *IUCN* dataset.

