# OpenReview forum: "Active Learning-Based Species Range Estimation"
_NeurIPS.cc/2023/Conference — NeurIPS 2023 poster_

### Official Review · Reviewer_RtuW · 2023-06-28

**Soundness:** 4 excellent
**Presentation:** 4 excellent
**Contribution:** 3 good
**Rating:** 6
**Confidence:** 4

**Summary:**

This paper tackles the problem of species distribution modeling from limited observation data. An active learning approach is proposed that makes the assumption that an unmapped species can be represented by a weighted combination of candidate species models (previously trained offline on a disjoint dataset). A sampling strategy is described that uses the aggregate uncertainty of the candidate models to select the next location to query. For representing location, a pre-existing feature representation is derived from a model trained on weakly labeled presence-only data.  Results on two benchmark datasets outperform baselines and an ablation study is presented.

**Strengths:**

- Addresses an important and interesting problem. Species distribution modeling has received a great deal of attention in the literature lately, especially related to fine-grained visual categorization. This work is of the first to explore how active learning learning can be applied to this problem to estimate range models from limited amounts of observation data.

- The proposed approach has technical novelty. The key idea is to take advantage of an existing set of species range models as a hypothesis set. During training, a range model for an unseen species is estimated by a weighted average of the candidate models. To capture fine-details, this candidate set is augmented with a model trained on the accumulated observations of the new species. In addition, a modified form of uncertainty sampling is also used which leverages the candidate models.

- Extensive evaluation, including an ablation study. The results indicate that design decisions directly lead to performance improvements, for example the hypothesis set sampling approach and the inclusion of the online model in the candidate set.

- Well written, easy to follow.

- Sufficient detail provided that the proposed method is likely reproducible.

**Weaknesses:**

- The proposed approach is a relatively straightforward application of active learning beyond the introduction of the candidate models.

- The approach is highly dependent on the hypothesis set (candidate models) being diverse such that a new species can be represented by some combination of them (L190). To partially address this, the authors add a model trained on the accumulated observations to the hypothesis set.

- The evaluation is largely limited to simple baselines in the form of
  a self-ablation (incrementally adding to an active learning baseline).  Only one recent paper is compared against (not active learning-based), with diverging results across the two benchmark datasets. That said, it appears there are no obvious other baselines to compare against.

**Questions:**

My initital rating is a weak accept. This paper is exceptionally well written and was fun to read. The use of active learning for species range modeling has real-world utility. The proposed approach, while simple, has novelty in its construction and the results improve over baselines. I think this paper will be of interest to the community.

Suggestions:

- An interesting byproduct of the proposed approach would be to show which species contributed (and their range models) the most in the weighted average.

- The annotations on Fig. 4 are hard to see without zooming in.

**Limitations:**

Limitations sufficiently addressed in the manuscript.

---

> ### Author Rebuttal · Authors · 2023-08-09
>
> **[RtuW-1] The proposed approach is a relatively straightforward application of active learning beyond the introduction of the candidate models.**
> Our approach consists of a novel exploration of active learning to the problem of efficiently estimating spatially varying quantities when there are informative priors available. It is made possible by the recent availability of large-scale citizen science collected species observation data and efficient neural network-based models for estimating their ranges. Our approach significantly outperforms conventional active learning baselines. We believe that the problem setting is unique, the application area is important, and the task will benefit from future investigation from the machine learning community.
>
> **[RtuW-2] The approach is highly dependent on the candidate hypothesis set being diverse such that a new species can be represented by some combination of them. To partially address this, the authors add a model trained on the accumulated observations to the hypothesis set.**
> Indeed, the diversity of the hypothesis set is an important part of our approach. In *Fig 3 (top right)* in the main paper we quantify the impact on downstream performance of significantly reducing the size of the hypothesis set. We observe that our approach is robust to large reductions and still outperforms the logistic regression random and uncertainty baselines even with very small numbers of candidate models. As noted in the question, the benefit of the added online trained model is most apparent when the candidate set only contains 1000 models or fewer. Adding this model to the candidate set is trivial and it has no meaningful impact on the amount of additional computation required.
>
> **[RtuW-3] The evaluation is largely limited to simple baselines in the form of a self-ablation (incrementally adding to an active learning baseline). That said, it appears there are no obvious other baselines to compare against.**
> In section C4 in the appendix we outline the twelve different baseline methods that we compare against. This does not include different ablated variants of our proposed model where we explore the impact of different feature spaces, i.e. *Fig 3 (top left)*. Given that we are one of the first works to explore the active learning setting of this problem, there are no other obvious baselines to compare against. We hope that our work, and the comprehensive experiments contained within, will serve as a valuable benchmark for this problem going forward.
>
> **[RtuW-4] An interesting byproduct of the proposed approach would be to show which species contributed (and their range models) the most in the weighted average.**
> This is a very interesting suggestion - thanks! We show an example of such as visualization in *Fig B3* of the rebuttal PDF. We will add additional improved images of this form to the final text to showcase these combinations across a diverse set of species.
>
> **[RtuW-5] The annotations on Fig. 4 are hard to see without zooming in.**
> We will make these larger. Thanks for pointing this out.

---

> > ### Comment · Reviewer_RtuW · 2023-08-15
> >
> > Thanks for the response. I have read the other reviews and all author rebuttals. As the majority of reviewers are in agreement, I will be retaining my initial rating.

---

### Official Review · Reviewer_yX8t · 2023-07-07

**Soundness:** 3 good
**Presentation:** 4 excellent
**Contribution:** 3 good
**Rating:** 7
**Confidence:** 3

**Summary:**

**Paper Summary**

This paper tackles the data-driven estimation of geographical habitat area for a given species; that is, given a certain geographic location (i.e., a "point on the map"), the goal is to predict how likely a certain species is present in this location. Prior efforts train models in an "offline" setting, assuming a finite training data set that indicates (for a fixed set of different species) the presence (or absence) of the species. Datasets that signal the presence (and possibly absence) of a large number of species are readily available.

This paper points out that dataset expansion for species not covered by existing datasets would be cumbersome- we would need to collect presence indication signals for new (or not yet recorded) species worldwide by possibly uniformly sampling presence indication evidence, which can then, in turn, be used to update/re-train existing models.

This paper investigates an interesting angle of this general problem: can we leverage *existing* (already collected) datasets to obtain a "weak" classifier for a new species, which can then guide the data collection efforts to regions, such that with as few samples we can decrease the overall presence entropy *as much as possible*?

This is akin to active learning, where the idea is to label as-informative-as-possible recorded data points. Different to that, in this case, the model suggests geographic regions in which we should sample the data for the new species (i.e., obtain presence/absence signals), such that the range estimates for new species will be as accurate as possible with as few data points (i.e., manual collections) as possible.

The overall idea behind the proposed work is to leverage the similarity between the (new) query species and existing species: this is based on the intuition that "similar species" (in terms of similarity in the feature space) will likely habitat similar territories. Such an approach can then be applied iteratively by incorporating newly-obtained presence/absence signals, suggesting increasingly more informative locations for the data collection.

The idea behind the approach is simple and intuitive: assuming an ensemble of (trained) linear models for a set of species, a model for a new species can be estimated as a weighted combination of existing linear models (where the model weight is proportional to the similarity in the feature space between the new (query) species and species database). This model then provides a "heatmap" over geographic locations, and guides the data collection -- new data points can be collected in the areas where the model is most uncertain about the new species' presence.

**Core Contributions/Insights**

The core contribution of this work is the insight that techniques similar to active learning can be used to streamline the data collection process. This is based on the key insight that existing datasets contain a sufficiently large number of species samples, such that a linear combination of already trained linear classifiers provide a reasonably good predictor for the presence of new species.

**Key Results**

The paper convincingly demonstrates that the proposed hypothesis ensembling outperforms logistic regression, and adding an online-trained model to the ensemble further improves the performance. I cannot comment on whether any relevant baselines are missing, as I am not actively working in species range estimation or active learning, and I am, therefore not familiar with related work. I hope my colleagues can comment on this.

Experimental validation also discusses the impact of the number of existing species (models) for model averaging on the final performance and establishes that the proposed approach is useful when a large number of specie presence models is readily available.

**Potential for Impact**

Specie habitat estimation has the potential to assist scientists in monitoring how the habitats of species are changing over time, which is especially important in the era in which global warming presents an existential threat to several species. Therefore, data-driven methods that can provide reasonable estimates based on (I assume difficult to obtain) evidence of species have important practical implications. As monitoring (and consequentially, data collection) is a continual process, developing models that can guide data collection efforts using active learning techniques makes sense.

**Rating Justification**

I think this paper presents an interesting, well-motivated, and well-executed work with direct practical relevance to monitoring species habitats. At this point, I see no obvious objections to why this paper should not be accepted.

This is a low-confidence review written by a reviewer entirely unfamiliar with bird-species range estimation and only weakly familiar with the field of active learning.

**Post Rebuttal Comments**

I want to thank the authors for their thorough responses. All my questions were addressed, and I see other reviewers are also positive about this paper. I am upgrading my rating to accept.

**Strengths:**

* Application relevance: this work has the potential to guide future data collection efforts for species range estimation. Monitoring species range is important for understanding how the earth's biosphere evolves under rapidly changing environmental conditions.
* Paper presentation: this paper is very well written and summarizes well both the application domain (species range estimation), as well as existing methods that are commonly being used, together with simple and intuitive taxonomy of prior efforts. Core contributions are presented clearly, well-motivated, and experimentally validated. As someone reading a paper on species range estimation for the first time, I really appreciate Sec. 3, which summarizes how this problem is typically tackled in the literature.
* Motivation: the paper makes a convincing case for investigating which areas should be used to collect data in the next training cycles using active learning techniques.
* Results: I find the paper's claims convincingly validated.


**Weaknesses:**

In addition to the weaknesses discussed in the limitation section, it would be fair to point out that the proposed approach is most likely *only* applicable to new species that are sufficiently similar to species for which data is readily available. I assume this approach would fail in the case of potentially newly-discovered species that would hypothetically be quite unlike anything that was observed before. As I am not familiar with the biological aspect of this task, I am not sure if this is a practical concern.

**Questions:**

Did the authors perform any kind of analysis on whether the intuition that most related species are a good indicator for the presence of new species holds? I.e., for a query species, are highly-weighted classifiers indeed representing most related species, and their presence is a good indicator? Did the authors observe any failure cases? For example, the presence of domestic cat, which is related to wild cats, is probably not a good presence indicator for, e.g., lion.

**Limitations:**

The paper thoroughly discusses the limitations in a designated section. I find this section insightful.

---

> ### Author Rebuttal · Authors · 2023-08-09
>
> **[yX8t-1] Is the proposed approach only applicable to new species that are sufficiently similar to species for which data is readily available?**
> If the candidate hypothesis set does not sufficiently cover the space of possible ranges for a held out test species of interest this will indeed impact performance. In *Fig 3 (top right)*, we explore a variant of this setting whereby we reduce the number of species present in the hypothesis set from 44,181 down to 100. As more species are removed, the amount of overlap with the test species is also reduced. However, we observe that our approach is relatively robust to large reductions in the number of species. Given that our full candidate set contains tens of thousands of species, we have good coverage of the types of ranges that might be encountered for a potential held-out species. As indicated in *Fig 3 (top right)*, it is reasonable to assume that as more species are added to the candidate set (e.g. there are several hundred thousand species on the iNaturalist website), performance may improve further. The requirement for a diverse set of species in the candidate set is briefly mentioned on line 328, but we agreed that this is an important point and we will emphasize this constraint in the final text.
>
> **[yX8t-2] Do highly-weighted classifiers indeed represent most related species, and is their presence a good indicator?**
> This is an interesting question. In *Fig B2* in the rebuttal PDF we illustrate the similarity between different species from the candidate set. In *Fig B3* in the rebuttal PDF, we illustrate an example of the weighted combination of the classifiers and show how they represent the test range of interest. We will include more visualizations of the weighted combinations in the final text.
>
> **[yX8t-3] Did the authors observe any failure cases? For example, the presence of domestic cat, which is related to wild cats, is probably not a good presence indicator for a lion.**
> In *Figs A5* and *A6* in the appendix we observe a histogram of the per-species performance for our approaches at different time steps. We observe that after 50 timesteps, there are only a relatively small number of species where the estimated range is poor. An example failure case is illustrated in *Fig A9* where our WA_HSS approach fails to query observations for the species in Africa. This failure could be explained by the relatively unusual required combination of European and Sub Saharan ranges for this particular species. However, it is worth noting that in the example in *Fig A9*, we only observe the model predictions up until the 8th time step, and it is very plausible that in later timesteps the performance will improve as a result of better samples, and thus a better model weighting, being obtained.
>
> In our setting, "relatedness" is computed based on the similarity between two species in the candidate hypothesis set which are in themselves indicative of range similarity, i.e. two species that have similar ranges will have similar weight vectors $\mathbf{\theta}_k$. As a result, taxonomic similarity between two species (as in the cat example provided in the question) is not actually taken into account or modeled explicitly and thus we would not be susceptible to the limitation outlined in the question. However, this is an interesting question and we will clarify the point about taxonomic relatedness in the updated text.
>
> We also discuss taxonomic relatedness in our response to **yFB1**. As a result of divergent evolution, species with common ancestors can evolve and accumulate differences, e.g. a preference for different types of habitats which can manifest as different ranges (e.g. African versus Indian Elephants). As a result, even if two species are taxonomically related (e.g. a domestic and wild cat) this information may not provide any useful signal for our range estimation task as visual appearance is not taken into account.

---

> > ### Comment · Reviewer_yX8t · 2023-08-12
> > **Thank you!**
> >
> > Thanks for the thorough response! I am happy with the rebuttal, and I didn't see any other major weaknesses raised by my colleagues. I upgraded my rating to accept.

---

### Official Review · Reviewer_B3F8 · 2023-07-09

**Soundness:** 4 excellent
**Presentation:** 4 excellent
**Contribution:** 2 fair
**Rating:** 4
**Confidence:** 4

**Summary:**

The paper explores a method for geographical range estimation of a species.  One of the primary challenges in range estimation is the scarcity and costliness of data collection, particularly for rare species. To address this issue, the paper introduces an active learning framework as a solution. The proposed method offers two key advantages. Firstly, it utilizes a joint feature representation that combines information from multiple species, enhancing the accuracy of the range estimation. Secondly, it employs an active learning scheme to efficiently collect and learn from data to estimate a geographical range of new species. The paper also proposes a linear combination of learned classifiers to develop a new classifier for each new species. The effectiveness of the proposed method is evaluated on two datasets: the IUCN dataset and the S&T dataset.

**Strengths:**

The paper is written clearly. The description of the problem setup and the technical approach is intuitive, allowing readers to grasp the concepts easily.

The paper makes a significant contribution to the field of species range estimation. It achieves a notable performance increment and thoroughly explores the problem with a comprehensive analysis.

Furthermore, the paper provides a comprehensive analysis of the proposed method, particularly focusing on the active learning scheme. This includes multiple stages such as initial representation learning, acquisition function, and model training. The authors thoroughly examine and analyze the proposed active learning framework, providing a comprehensive understanding for each components.


**Weaknesses:**

While the paper thoroughly analyzes its proposed method and provides empirical results for the specific problem of range estimation, it does not explicitly highlight the unique machine learning challenges of this problem or how they connect to the technical contribution of the paper. Range estimation can be seen as sharing commonalities with other common machine learning problem settings. For instance, the presence-absence data setting can be considered a supervised classification problem, while the presence-only data setting resembles one-class learning or semi-supervised learning, where the labeled set only contains positive samples.

The paper primarily focuses on the application aspect by developing an efficient active learning framework for the range estimation problem. However, it does not provide a detailed explanation of why the chosen solution, which employs widely used machine learning techniques, was devised among numerous alternative approaches. The paper would benefit from explicitly stating the unique machine learning challenges posed by the range estimation problem and how the proposed method effectively addresses those challenges. Additionally, a more comprehensive discussion of the motivation behind the chosen approach would enhance the clarity and understanding of the paper.


**Questions:**

What is the unique machine learning challenge of the range estimation problem that distinguishes it from other machine learning problems or applications?

Equation (3) averages the classifier parameters. How does the model perform if averaging is applied to the output rather than to the parameters? Why did the authors decide to apply the average on parameters rather than output?

Equation (5) uses average over outputs while equation (3) averages over parameters. Why should uncertainty computation (Eq. 5) be different from inference computation (Eq. 3)? Is there an intuition behind this?

What happens if the pretrained classifiers are not used, but instead train a new species classifier from scratch on S_t? This is an important baseline but is missing in the paper.

It is interesting that nn_rand performs way better than the other two features; loc_feats and env_feats. What would be the reason for this? Is nn_rand trained on S_t? Or is nn_rand fixed as the random initial weights?


**Limitations:**

The paper addresses the limitations of the proposed method and its broader impacts.

---

> ### Author Rebuttal · Authors · 2023-08-09
>
> **[B3F8-1] The paper does not explicitly highlight the unique machine learning challenges of this problem or how they connect to the technical contribution of the paper.**
> As noted on line 118, when presence and absence data is available, the range estimation problem for a single species can be viewed as a binary classification task. The core machine learning task explored in our work is how to estimate a species’ range while sampling the fewest number of locations (see Sec 3.2). While not the focus of this work, the training of the backbone neural network feature extractor from presence-only data can be viewed as a single positive multi-label problem (see [11]). As noted on line 161, our setting differs from conventional binary classification (e.g. image classification) as the ranges of different species are not necessarily distinct, but can instead have a potentially large amount of overlap. This observation is what makes our proposed solution possible. We will update the text to explicitly state the core machine learning task explored and precisely indicate how our proposed solution addresses it.
>
> **[B3F8-2] A more comprehensive discussion of the motivation behind the chosen approach would enhance the clarity and understanding of the paper.**
> The motivation behind our approach is illustrated in *Fig 1*. The main assumption is that given a sufficiently diverse set of candidate species range estimation models it should be possible to describe the range of a previously unseen test species. The intuition is that by leveraging this prior set of possible predictions we are able more rapidly estimate a species range compared to learning from scratch or any of the numerous other baseline we compare to. In *Fig 3 (top right)* we show that as we reduce the number of candidate models available, performance on held-out species decreases. We will update the text to better explain why the chosen solution was devised.
>
> **[B3F8-3] Eqn. 3 averages the classifier parameters. How does the model perform if averaging is applied to the output rather than to the parameters? Why did the authors decide to apply the average on parameters rather than output?**
> By averaging the classifier parameters we obtain a compact 256 dimensional encoding of a species’ global range. Our neural network feature extractor is trained on 44,181 species (line 227). If we instead averaged the output of this model, it would significantly increase the amount of computation required as this scales with the number of species (which could be up to the order of millions of species as opposed to the fixed dimensionality of the network’s features). In addition, we would not have a compact representation of the classifier for the new species once active learning is complete and would instead have to store and evaluate all 44,181 classifiers to estimate the new species’ range.
>
> **[B3F8-4] Eqn. 5 uses average over outputs while eqn. 3 averages over parameters. Why should uncertainty computation (eqn. 5) be different from inference computation (eqn. 3)?**
> Eqn. 3 updates the model parameters based on the agreement with the data currently observed and eqn. 5 selects the next location to be sampled that is most uncertain according to the set of candidate models. The intuition here is that we would like to select geographical locations that allow us to downweight the largest number of irrelevant candidate models at each step. In doing so, we are able to focus in on the candidate models that have the most agreement with the observed data and then efficiently represent the learned classifier as a single weight vector.
>
> **[B3F8-5] What happens if the pretrained classifiers are not used, but instead train a new species classifier from scratch on $S^t$?**
> To clarify, the logistic regression baselines in *Fig 2* and *Table A1* are already trained from scratch on $S^t$ for each species. They make use of the pretrained feature space (line 123) but do not use the pretrained candidate models. Unsurprisingly, without the backbone neural network features, these models perform very badly as the raw location or environment features are not very informative (see *Fig B1* in the rebuttal PDF where we train the requested classifiers on S_t directly using different features). We observe similar poor performance for the *WA_HSS+* model in *Fig 3 (top left)* when not using the network’s features.
>
> **[B3F8-6] Why does nn_rand perform much better than loc_feats and env_feats (Fig 3 - top left)?**
> In *Fig 3 (top left)*, as in all our experiments, the feature space remains fixed during active learning (i.e. we do not update the neural network) and only the species classifier weights are trained (line 231).  The *nn rand feats* come from a randomly initialized but untrained version of the standard network (line 285). In Fig 3 (top left) we see that these features perform much worse than the trained network features (*nn env feats* or *nn loc feats*). However, this fixed random higher dimensional projection of the input coordinates performs much better than only using the low dimensional location or environmental features (*env feats* or *loc feats*). This is because the low dimensional features are not sufficiently linearly separable, but the higher dimensional fixed random projections are. We will add this discussion to the revised text.

---

> > ### Comment · Reviewer_B3F8 · 2023-08-18
> > **Thank you for the clarification.**
> >
> > Thank you for clarifying the questions.
> >
> > 1. Regarding the rand_nn experiment in Figure 3, how is nn_rand feat obtained?
> > Is nn_rand feat output of an untrained neural network with location input? (or env input or env+location?)
> >
> > 2. [B3F8-3] The computational trade-off between parameter averaging and output averaging comes from the size of observed samples versus (cost for parameter averaging) the size of the unexplored area, i.e., the number of location samples to be predicted for species existence (cost for output averaging). A brief discussion of the computational cost would enhance the readability of the paper. Please correct me if there is any misunderstanding on my side.
> >
> > 3. [B3F8-5] Thank you for the clarification but I am still unclear how the classifier trained from scratch performs.
> > As I understand, the feature extractor is frozen, so classifiers are only updated. Three parts are introduced in this paper, 1) linear combination of hypothesis models using equation (3), 2) Active sampling using equation (5), 3) online training with an updated set of annotated samples S^t. I wonder how the model would perform with only 2) and 3) without 1). In other words, how would a new species classifier perform when trained from scratch but using active sampling and online training? This is not a critical question, but I would like to know the importance of training from linearly combined weights instead of starting from scratch.

---

> > > ### Author Response · Authors · 2023-08-20
> > > **response to reviewer**
> > >
> > > Thank you for your responses and for your engagement with our paper.
> > >
> > > **Q1 - [B3F8-6]** Your first explanation is correct. “nn rand feats” are the output of an untrained neural network that takes *location* as input. We will clarify this in the paper.
> > >
> > > **Q2 - [B3F8-3]** We agree with your comments and interpretation of the computational cost. We will add a discussion of this to the final paper so that readers can better understand the merits of our approach, e.g. the benefits of the compact representation stemming from eqn 3, as noted in our previous response. Thanks for the suggestion.
> > >
> > > **Q3 - [B3F8-5]** To clarify, the feature extractor is frozen in all experiments (with the exception of the AN_FULL_E2E baseline). This is motivated by two primary reasons: (i) the amount of labeled data that is obtained during the early stages of active learning is very small, i.e. just a few samples, whereas the backbone neural network has multiple layers and (ii) it is desirable for the model update step to be efficient so that it does not become a bottleneck between the active sampling steps. Point (i) is particularly important, as it would not be trivial to select the appropriate training hyperparameters for each time step when training end-to-end with limited data.
> > >
> > > On line 258 we describe a baseline that we believe is close to what you request, i.e. the AN_FULL_E2E backbone is trained in an end-to-end manner with all the labeled data. As can be observed in Fig 2, we outperform this strong baseline for the IUCN dataset and are approaching it in the case of S&T. This gives a sense of the impact of training end-to-end versus using a linear combination of weights. Furthermore, Fig 3 (top right) illustrates the performance decrease of our method when the linear combination is not a good approximation for a test species’ range owning to a reduced candidate set size. Also related is LR_HSS in Fig 2 which utilizes eqn 5 for active sampling and has online training (of just the classifier) using S^t, but does not generate a model using eqn 3. In this case sampling locations are determined by the weighted hypothesis set, but unlike WA_HSS, the classifier is trained on S^t using standard logistic regression (i.e. from scratch). This model performs poorly on both the IUCN and S&T datasets.
> > >
> > > We agree that yours is an interesting question, and we will further address it in the final text. Also, please do not hesitate to follow up with any additional questions if we have not fully answered your questions or if we have misinterpreted them.

---

### Official Review · Reviewer_yFB1 · 2023-07-13

**Soundness:** 3 good
**Presentation:** 3 good
**Contribution:** 2 fair
**Rating:** 6
**Confidence:** 3

**Summary:**

This paper proposes an active learning framework for estimating the geographical range of species from sparse observations. The core idea is to exploit a large set of pretrained range estimation models from many different species. This paper also provides extensive experiments including detailed ablation studies and comparisons with baselines, demonstrating the effectiveness of the proposed active learning approach.

**Strengths:**

1. It is technically sound to exploit a large set of pretrained range estimation models from diverse species to reduce the required observations for training a new model.
2. This paper provides detailed ablation studies and comparisons with existing methods and baselines and convincingly demonstrates superior performance.
3. This paper is well-written and easy to follow.

**Weaknesses:**

1. The technical novelty is limited. It is reasonable yet straightforward to exploit a set of pretrained range estimation models for training a model for a target species.
2. The exact reason why it is meaningful to learn a linear combination of pretrained range estimation models needs further clarification. Is this because the range of different species is highly correlated and the learned weights indicate some correlation among different species, or such a linear combination indicates some general habitat candidates? A baseline probably stronger than the LR_random (Fig. 2) is to train a general classifier on joint presence data from all available species. Such a general model may indicate whether a given location is a good habitat for birds or other animals.
3. Is it visualize the weights in the linear combination for some target species? Do the weight visualizations indicate any correlations among species? If so, is such a correlation related to species taxonomy? It might also be useful to incorporate the species taxonomy as input to the model.

**Questions:**

1. Could you elaborate on the details of LR_HSS? For HSS (hypothesis set selection)? If logistic regression instead of weighted averaging is applied during training, how to apply HSS (Eqn. 5) at the test time? It looks like the weights per hypothesis H is required during active querying.
2. Typos: #199-#200, WA_HHS should be WA_HSS.

**Limitations:**

1. Limited technical novelty. The core idea of this paper is to exploit a set of pretrained range estimation models to reduce the number of observations required for training a new model for target species. This problem studied in this work is very specific to the range estimation problem.
2. A stronger baseline that trains a general classifier should be considered as described above.
3. The role of weighted averaging needs more clarification.

---

> ### Author Rebuttal · Authors · 2023-08-09
>
> **[yFB1-1]  The technical novelty is limited. It is reasonable, yet straightforward, to exploit a set of pretrained range estimation models for training a model for a target species.**
> Respectfully, we disagree. To the best of our knowledge, we are the first work to show that it is possible to make use of a large set of pretrained models to perform efficient active learning in the context of spatially varying quantities. We quantitatively show that our approach is superior to over ten different active learning baselines (see section C4 for a description of the baselines) on the task of species range estimation across two different datasets. The problem of efficiently mapping species ranges is an important one for the ecological and conservation applications which we believe needs more attention from the machine learning community. Our problem setting is distinct from seemingly related tasks (e.g. active learning of image classifiers) where leveraging such a prior is not meaningful as the potential similarity between categories is not as useful of a signal.
>
> **[yFB1-2] Why is it meaningful to learn a linear combination of pretrained range estimation models?**
> Our solution is motivated by the observation that “the ranges of different species are not necessarily distinct, but can instead have a potentially large amount of overlap” (line 161). The intuition behind this assumption is visualized in *Fig 1* where we illustrate that species that are similar in weight space have similar ranges (see also rebuttal Fig B2). Given a sufficiently diverse set of possible pretrained range estimation models, it is thus reasonable to assume that some combination of those ranges will be a good proxy for representing the range of a previously unseen test species. As noted on line 122, we can use a set of learned features extracted from a jointly trained geospatial deep neural network as our feature vector for each geographical location (which can be viewed as a learned encoding of the local “habitat”). The range for a species of interest is then predicted as the dot product between these neural network extracted features and the learned weight vector for that species (line 127). To further emphasize the impact of the set of pretrained range estimation models, in *Fig 3 (top right)* we observe that the performance of our approach drops when we reduce the number of species in the candidate set of pretrained range estimation models. To provide additional motivation as to how the linear combination works, in *Fig B3* in the rebuttal PDF we illustrate an example learned combination. We will add additional visualizations with an associated discussion to the final text.
>
> **[yFB1-3] A baseline stronger than the LR_random would be to train a general classifier on joint presence data from all available species.**
> Apologies, if we have misunderstood your request, but we believe that your suggestion is already included in the paper. To clarify, *LR_random* uses the same neural network backbone 256 dimensional feature space (line 228-232) as our model. However, it differs in how active sampling is performed (here random selection) and how the classifier is obtained for a test species (it uses the conventional binary cross entropy training objective). In *Fig 2* we can see that this model does not perform very well. A stronger baseline is the *LR_uncertain* model. In this instance, the location selection is performed using uncertainty sampling. Thus both of these baselines make use of the rich and informative feature space learned from the neural network that has been trained jointly on the large set of species observation data. We will update the descriptions of the baselines in section C4 in the appendix to clarify this. If we have misunderstood your question, please let us know in the discussion phase and we will happily run any necessary additional experiments.
>
> **[yFB1-4] Does the linear combination of weights indicate any correlations among species? If so, is such a correlation related to species taxonomy?**
> Species that have similar learned weight vectors ($\mathbf{\theta}$ have similar predicted ranges. To explain a range for a new species, we make the assumption that this can be achieved by estimating a weighted combination of existing species weight vectors (Equation 3). In *Fig B3* in the rebuttal PDF, we illustrate an example of this weighted combination. We will expand on this discussion in the final text. Please see our discussion of taxonomy in the next question.
>
> **[yFB1-5] Would it be useful to use the species taxonomy as an input to the model?**
> This is an interesting question. Currently our approach does not encode or require domain specific information such as taxonomic relatedness. However, it is not clear if taxonomic information would be helpful. As a result of divergent evolution, species with common ancestors can evolve and accumulate differences, e.g. a preference for different types of habitats which can manifest as different ranges (e.g. African versus Indian Elephants). As a result, even if two species are taxonomically related, this information may not provide any useful signal for our range estimation task (even though taxonomic information would likely be helpful for image classification). It is possible that other forms of domain knowledge could be encoded into our model, but we leave this for future work.
>
> **[yFB1-6] Could you elaborate on the details of LR_HSS?**
> As noted on line 133 in the appendix, for this model, the next location to sample is selected based on the weighted hypothesis set and then the estimated range model is updated using binary cross entropy loss. This results in a single weight vector that can be applied at test time using the logistic model.  We will clarify this description in the final text.
>
> Thanks for pointing out the typo, we have fixed it.

---

> > ### Comment · Reviewer_yFB1 · 2023-08-18
> > **Follow-up**
> >
> > Thanks for the detailed clarification! Upon reading the rebuttal and other reviewers' comments, I'll raise my rating from borderline accept to weak accept.

---

### Official Review · Reviewer_WCfR · 2023-07-14

**Soundness:** 4 excellent
**Presentation:** 4 excellent
**Contribution:** 3 good
**Rating:** 6
**Confidence:** 4

**Summary:**

This paper proposes to:
1) Use crowdsourced data to obtain species range maps for a large set of data-rich species, and then
2) Learn a linear combination of these maps to represent the range of other, data-poor, species, while minimizing the number of required presence/absence queries via active learning.

**Strengths:**

The paper is well motivated, well structured, reads well and provides a good overview of the existing literature.
The proposed method is simple and intuitive, and the performance improvements quite substantial.

**Weaknesses:**

Although I'm quite positive about the paper, I do have some small doubts about whether a more general context placement would be required for a better fir in NeurIPS. I'm no expert, but it seems to me that the authors could connect the work to methods beyond species range estimation in order to connect better with the ML community. After all, the proposed method seems to perform reconstruction with a predefined basis. Otherwise, it seems like a paper more oriented exclusively towards the SDM community.

**Questions:**

As reflected in the weaknesses and limitations sections:
1) What other fields of application and methods in the ML community would have connections to the presented methodology?
2) Would this method eventually be feasible in reality?

**Limitations:**

Most limitations are properly addressed. The only thing I missed would be the feasibility of obtaining presence/absence queries. Is this something that has been done? Obtaining a presence given that the species is present is one thig, but obtaining an absence could be very challenging. Would this method eventually be feasible in reality?

---

> ### Author Rebuttal · Authors · 2023-08-09
>
> **[WCfR-1] What other application areas could your method be applied to and is this a good fit for NeurIPS?**
> This is an interesting question. Our approach is applicable to any spatially varying quantity where it is possible to obtain a hypothesis set over possible related predictions. Potential applications include sensor placement, geographical priors for image classification, and disease modeling, in addition to related tasks such as active data collection for training image classifiers. We will add a discussion of other application areas to the text. However, it is worth noting that the percentage of species, of the total known to science, with documented ranges is very small. As a result, there are a large number of unmapped species that methods such as ours, and future advancements, could be applied to. While the task of species range estimation has achieved attention from the machine learning community (e.g. *Phillips et al. ICML 2004*, *Chen et al. ICML 2018*, and *Cole et al. 2023*), our work lays the groundwork for future progress on the active learning version of the task which has not been explored to date.
>
> **[WCfR-2] Confirming if a species is present is easy. How difficult is it to obtain absence information, i.e. to confirm that it is absent for a particular location?**
> Platforms such as iNaturalist only record presence observations, however other platforms such as eBird ask users to upload both presence and absence observations. To date, eBird users have uploaded approximately 100 million checklists where each checklist contains information about the presence and absences of all bird species for the location they sampled. Thus this type of presence and absence data is readily available and often collected.
>
> It is true that confirming absences (true negatives) may be more time consuming than presences (true positives) for some species. However, the benefit of our approach is that we can significantly reduce the number of locations that need to be sampled to obtain an estimate of a species' range (see *Fig 2*). Furthermore, our approach is robust to noise in the form of false negatives (i.e. incorrectly stating that something is absent) that might arise in the observation process due to the potential difficulty of obtaining reliable absences (see *Fig 3 (bottom left)* and *Fig A1* in the appendix). Importantly, our hypothesis candidate set (line 164) is generated from a model that is only trained on presence data which can easily be obtained in abundance. However, we agree that it is important to point out the distinction between presence and absence observations and so will add a discussion of this in the updated text.

---

> > ### Comment · Reviewer_WCfR · 2023-08-14
> >
> > I appreciate the authors' response, as I think it would be important to touch these two aspects in the paper.

---

> > > ### Author Response · Authors · 2023-08-20
> > >
> > > We agree, we are very happy to add this - thanks for the suggestion. These will be easy additions to make to the final paper and will potentially even motivate future work in other related problem domains.

---

### Author Rebuttal · Authors · 2023-08-09

We thank the reviewers for their constructive comments and insightful questions. We appreciate the detailed reviews and words of support regarding the overall quality and clarity of the paper provided by each of the five reviewers. As there are no significant common concerns from the reviewers, we respond to their comments individually. We also provide three additional experiments/visualizations in the rebuttal PDF in response to specific reviewer comments. Finally, if there are any questions that are still not resolved via our replies, we are very happy to provide additional responses during the author/reviewer discussion period.

Only relatively minor changes in the text are requested to better motivate the problem setting, provide additional baselines that we have already added to the rebuttal PDF, justify some of the choices made, and draw connections to other related machine learning tasks. These changes are straightforward to implement and will further enhance the overall clarity of the paper.

---

### Decision · Program_Chairs · 2023-09-21

**Decision:**

Accept (poster)

**Comment:**

This paper considers the problem of learning species distributions from observations, and how best to direct limited ecological monitoring efforts. The authors propose an active learning method to identify high-priority locations for gaining additional data, by using the distributions inferred for some species to feed into a mixture of experts informing the potential distributions of others. While there was some minor disagreement on this paper during review, there was general consensus that the method proposed was a valuable ML contribution in addressing an important problem. The biggest concern raised was in "novelty", but I concur with several of the reviewers in seeing this paper as an effective example of application-guided innovation, where the methodological advances were appropriately motivated by application-specific challenges and criteria for success. Accordingly, I recommend the paper strongly for acceptance.